# Microglia innately develop within cerebral organoids

Paul R. Ormel[1,2], Renata Vieira de Sá[1], Emma J. van Bodegraven[1], Henk Karst[1], Oliver Harschnitz [1,3], Marjolein A.M. Sneeboer[1,2], Lill Eva Johansen[1,3], Roland E. van Dijk[1], Nicky Scheefhals[4], Amber Berdenis van Berlekom[1,2], Eduardo Ribes Martínez[1], Sandra Kling[1], Harold D. MacGillavry [4], Leonard H. van den Berg[3], René S. Kahn[2], Elly M. Hol[1,5], Lot D. de Witte[1,2] & R. Jeroen Pasterkamp [1]

Cerebral organoids are 3D stem cell-derived models that can be utilized to study the human brain. The current consensus is that cerebral organoids consist of cells derived from the neuroectodermal lineage. This limits their value and applicability, as mesodermal-derived microglia are important players in neural development and disease. Remarkably, here we show that microglia can innately develop within a cerebral organoid model and display their characteristic ramified morphology. The transcriptome and response to inflammatory stimulation of these organoid-grown microglia closely mimic the transcriptome and response of adult microglia acutely isolated from post mortem human brain tissue. In addition, organoid-grown microglia mediate phagocytosis and synaptic material is detected inside them. In all, our study characterizes a microglia-containing organoid model that represents a valuable tool for studying the interplay between microglia, macroglia, and neurons in human brain development and disease.

[1] Department of Translational Neuroscience, Brain Center Rudolf Magnus, University Medical Center Utrecht (BCRM-UMCU), Utrecht University, P.O. Box 85500, Universiteitsweg 100, 3584 CG Utrecht, The Netherlands. [2] Department of Psychiatry, BCRM-UMCU, Utrecht University, P.O. Box 85500, Heidelberglaan 100, 3508 GA Utrecht, The Netherlands. [3] Department of Neurology, BCRM-UMCU, Utrecht University, P.O. Box 85500, Heidelberglaan 100, 3508 GA Utrecht, The Netherlands. [4] Cell Biology, Department of Biology, Faculty of Science, Utrecht University, Padualaan 8, 3584 CH Utrecht, The Netherlands. [5] Department of Neuroimmunology, Netherlands Institute for Neuroscience, An Institute of the Royal Netherlands Academy of Arts and Sciences, Meibergdreef 47, 1105 BA Amsterdam, The Netherlands. These authors contributed equally: Paul R. Ormel, Renata Vieira de Sá. These authors jointly supervised this work: Elly M. Hol, Lot D. de Witte, R. Jeroen Pasterkamp. Correspondence and requests for materials should be addressed to L.Witte. (email: L.D.deWitte@umcutrecht.nl)

Microglia are the resident innate immune cells of the central nervous system (CNS). While microglia originate from the mesoderm lineage, other CNS cells like neurons and astrocytes are derived from neuroectodermal progenitors[1,2]. Besides their immune functions, microglia also control the number of neuronal precursor cells, synapse formation, and synapse elimination[3–6]. Impaired interaction between microglia, neurons, and astrocytes is increasingly linked to neurodegenerative and neurodevelopmental disorders, such as Alzheimer's disease, amyotrophic lateral sclerosis (ALS), autism, and schizophrenia[7–10]. The majority of studies on neuron-glia interactions are performed in rodents, due to a lack of adequate human model systems that recapitulate the development of microglia in vivo and the interplay between microglia, macroglia, and neurons in a 3D context. Human model systems, however, will be vital to understand how neuron-glia interactions impact human CNS development, physiology, and pathology.

Recently, stem cell-derived organoid models offer the possibility to study cellular development and inter-cellular interactions within a 3D human brain microenvironment[11]. Organoids are generated by culturing embryonic stem cells (ESCs) or induced pluripotent stem cells (iPSCs) into embryoid bodies with the potential to develop progenitors from all the three germinal layers: endoderm, ectoderm, and mesoderm. The embryoid body is pushed into a specific tissue fate, like CNS, by complementing the cell culture medium with growth factors and/or inhibitors. Due to their self-organizing capacity, cell aggregates will then develop into CNS organoids consisting of neuronal sub-types and macroglia, forming specialized CNS-areas such as cortex, hippocampus, and retina[11–13]. A reported limitation of CNS organoid protocols is that they drive cells into the neuroectoderm lineage by inhibiting mesoderm and endoderm formation. Consequently, CNS organoids have been suggested to lack the complete combination of cells derived from different germ layers that are present in the brain in vivo, including microglia[11,14].

Dual-SMAD inhibition is commonly used to quickly induce neuroectoderm formation in directed differentiation protocols[15]. However, Lancaster and colleagues published a protocol to generate cerebral organoids without the use of any inhibitors or molecular pathway manipulators[12,16]. This approach led to the first 3D organoid model containing distinct brain regions like hippocampus, retina, and different cortical domains[12,16]. Their study reports that non-neuronal cell types are expelled from inside the organoids upon matrigel embedment and that the remaining cells are from the neuronal lineage[17]. The lack of dual-SMAD inhibition in cerebral organoid generation might explain why Quadrato et al.[18] recently reported the presence of mesoderm-derived progenitors in this model. We hypothesized that these mesodermal progenitors are able to differentiate into mature microglia instructed by the CNS microenvironment provided by neuroectodermal cells. Our results show that cells with a typical microglia molecular phenotype, morphology, and function are present in human cerebral organoids. This 3D organoid model in which microglia, macroglia, and neurons are present is important for studying microglia development, but also for studying neuron-glia interactions in human brain development and disease.

## Results

### Cerebral organoids contain progenitors from all germ layers.
Cerebral organoids were generated from human iPSCs according to the protocol described by Lancaster et al.[17] with some minor modifications (Fig. 1a and Supplementary Table 1). Two of the three iPSC lines used for the organoid cultures have previously been described by us (iPSC 1, 3)[19] and iPSC 5 was similarly generated and characterized (Supplementary Fig. 1a–h; Table 1). Ectodermal (PAX6), mesodermal (brachyury), and endodermal (AFP) progenitors were present at an early stage of organoid development (day 17; Fig. 1b). The presence of neuronal architecture and of astrocytes was confirmed by using a range of markers at day 31 (Supplementary Fig. 1i–p). Cleaved-caspase 3 immunostaining at several timepoints and batches (iPSC 1, 3, and 5; at day 17, 38, 52, and 66) showed constant sparse cell death in the organoids.

### Mesodermal progenitors develop into microglia-like cells.
Next, temporal expression of the mesodermal marker brachyury and the classical microglia marker IBA-1 was examined to assess if the mesodermal cells differentiate into microglia. Brachyury+ cells were present on day 17, 24, and no longer detected at day 52. IBA-1+ cells were detectable in low numbers on day 24 and had populated the organoids at day 52 (Fig. 1c). This transition took place around day 24, as at this timepoint a population of cells co-expressed both IBA-1 and brachyury (Fig. 1d). IBA-1+ cells were initially present in clusters distributed at specific sites in the organoid (day 31; Fig. 1e), but at later stages were found throughout the organoid (day 52; Fig. 1e). The fraction IBA-1+ cells within organoids showed limited variation between batches, timepoints, and donors (Supplementary Fig. 2a–c), and microglia quantity was positively correlated with TUJ1 expression (Supplementary Fig. 2d). Furthermore, microglia-containing organoids demonstrated proper neuronal development and cortical layer organization, based on expression of NEUN, PAX6, CTIP2, and TBR1 (Supplementary Fig. 2e). In rodents, microglia have a rounded morphology (amoeboid) at the moment they invade the brain parenchyma. At later stages, these cells develop characteristic ramifications. This process of differentiation is mirrored in the human organoids. IBA-1+ cells had a round morphology up to day 31, and showed a significant increase (Mann–Whitney $U = 0$, $p = 0.03$) in ramifications by day 52, here shown by an increase in perimeter (Fig. 1f, g). Microglial complexity progressed in time as IBA-1+ cells showed further ramification at day 66 (Fig. 1h).

### Microglia-specific molecular signature in cerebral organoids.
To determine whether the microglia within cerebral organoids developed in analogy to microglia in vivo, we assessed the expression of genes crucial for microglia development and function. RUNX1, a transcription factor involved in microglial development[20], was first expressed at an early stage of organoid development around the time of matrigel embedment (day 13), as were factors known to drive microglia development in vivo (*IL34*, *CSF1*, and *TGFB1*) (Fig. 2a). *RUNX1* regulates the microglia development genes *SPI1* (PU.1) and *CSF1R*[20], which we found expressed at day 24 (Fig. 2a). At this timepoint a clear increase in expression of other classical microglia markers was observed, i.e., *AIF1*/IBA-1, *CD68*, *ITGAM*/CD11b (Fig. 2a), *IRF8*, *TGFBR1*, *TGFBR2*, *TREM2*, *CX3CR1*, *HLADRA*, *C1QA*, and *PTPRC*/CD45[21–23] (Supplementary Fig. 3a). Microglial identity was further supported by complete co-expression of IBA-1 with the nuclear marker PU.1 (Fig. 2b) and major overlap with the expression of the microglia/macrophage marker CD68 (Fig. 2c) at day 52.

Taken together our results suggest that mesodermal progenitors within the cerebral organoids can develop into cells with a microglial phenotype, here referred to as organoid-grown microglia (oMG).

### The transcriptomes of oMG and adult microglia are similar.
To further validate the microglia phenotype of oMG, we performed

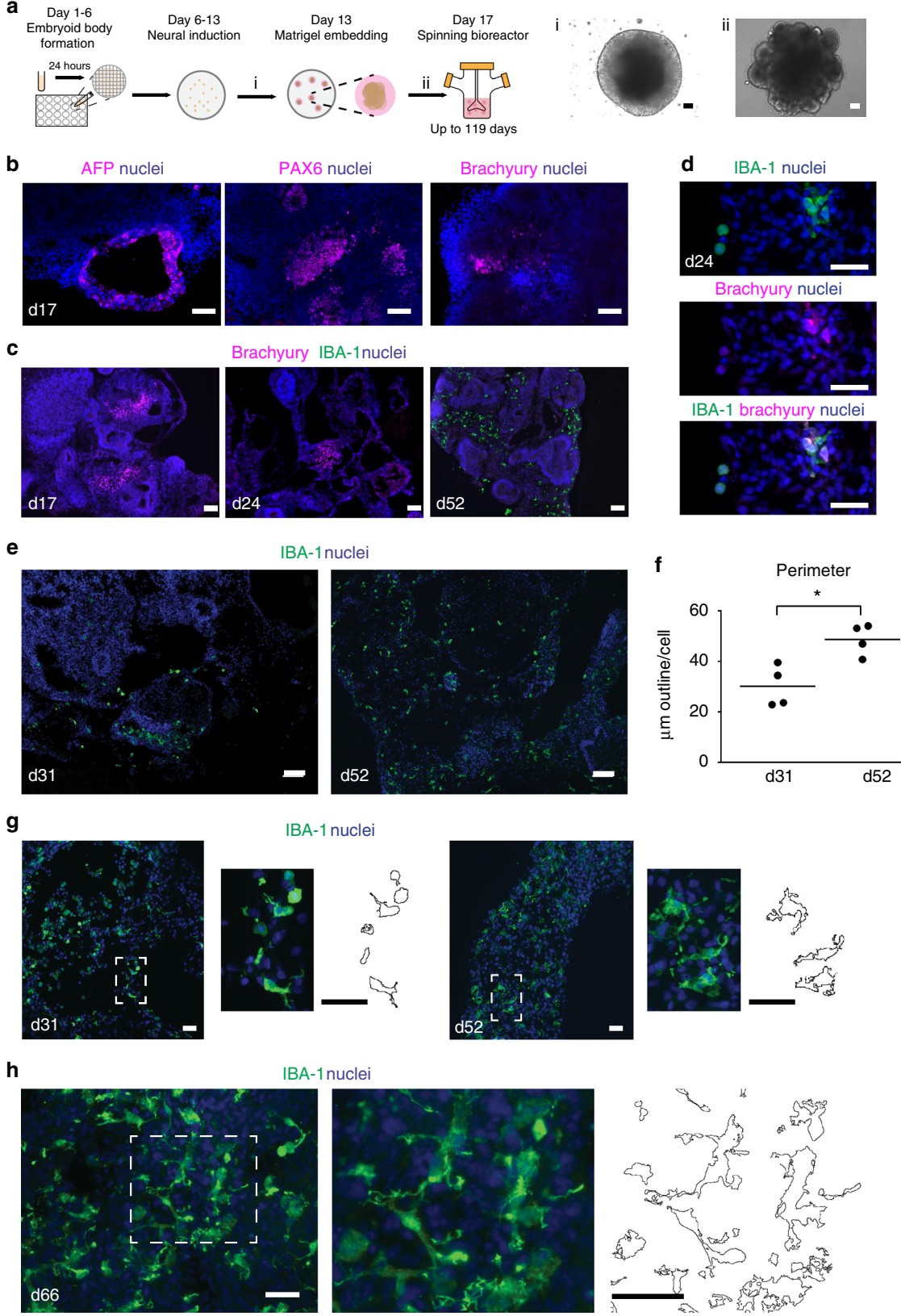

RNAseq transcriptomic profiling of oMG isolated from three independent organoid donors (oMG1, 3 and 5) at day 38 and day 52. Inter-donor variability was small (oMG $\rho = 0.91 \pm 0.001$) when examined with Spearman correlation coefficients (Supplementary Fig. 4a). Unsupervised hierarchical clustering shows that oMG clustered separately from adult microglia (adult MG1) (Supplementary Fig. 4b). When comparing the data with the original iPSC lines and fibroblasts from the same donors,

**Fig. 1** Mesodermal progenitors develop into microglia-like cells within cerebral organoids. **a** Schematic overview of the cerebral organoid protocol depicting the essential steps in the differentiation process. Embryoid bodies are formed (day 1–6) after which neuroectoderm is induced (day 6–13). Matrigel embedment provides an extracellular matrix to further grow and develop. Four days later they are transferred to a spinning bioreactor. Before matrigel embedment they have a smooth surface (i), which changes into a surface showing typical budding of the organoid 4 days after matrigel embedment (ii). Scale bar 100 μm. See also Supplementary Fig. 1, 2, and Supplementary Table 1. **b** AFP, PAX6, and brachyury immunostainings, which are markers for the germ layers endoderm ectoderm and mesoderm, respectively. Representative pictures of cerebral organoids are shown at an early stage of organoid development (day 17). Scale bar 40 μm. **c, d** Brachyury and IBA-1 immunostainings in cerebral organoids at 17 days (**c**), 24 days (**c, d**) and 52 days (**c**) in culture. Co-expression of IBA-1 and brachyury was visible at day 24. Scale bars 100 (**c**) and 40 μm (**d**). **e** IBA-1 immunostainings show distribution of microglia-like cells in cerebral organoids at day 31 and day 52 in culture. Scale bar 40 μm. **f, g** The perimeter of microglia at day 31 and day 52 quantified (**f**) with an automated approach by FIJI of IBA-1$^+$ immunostainings of cerebral organoids (**g**). $n = 4$ images of organoids quantified per timepoint, Mann–Whitney test $U = 0$, $p = 0.029$. Data is represented as median. Scale bars 40 μm. (close-up and perimeter are shown) (*$p < 0.05$). **h** Morphology of microglia-like cells illustrated by IBA-1$^+$ immunostainings after 66 days in culture. Scale bar 40 μm. (close-up and perimeter are shown) Representative pictures of cerebral organoids from iPSC 1 are shown

**Table 1 Clinicopathological data of the fresh human brain tissue and iPSC donors including: donor number, sex, age, pH CSF, post mortem delay, tissue or cell type, clinical diagnosis, and for which experiments the tissue was used**

| Donor | Sex | Age | pH csf | Post mortem delay | Tissue /cell type | Clinical diagnosis | Experiments |
|---|---|---|---|---|---|---|---|
| MG 1.1 16-137 | M | 77 | 6.46 | 12:45 | MFG, STG, THA | Control | RNAseq, Flow cytometry staining |
| MG 1.2 17-003 | F | 96 | 6.71 | 6:15 | MFG | Control | RNAseq |
| MG 1.3 17-005 | F | 60 | 7.07 | 5:30 | MFG | Control | RNAseq |
| MG 1.4 16-110 | F | 69 | 6.28 | 16:10 | MFG, SVZ | Depression | MACs, Stimulation |
| MG 1.5 17-017 | F | 67 | 7.07 | 8:40 | MFG | Depression | MACs, stimulation |
| MG 1.6 17-029 | F | 23 | N/A | 8:35 | SVZ | Depression | MACs, stimulation |
| MG 1.7 18-037 | M | 63 | 6.63 | 05:55 | SVZ | Parkinson's Disease | Phagocytosis |
| MG 1.8 18-039 | F | 82 | 6.60 | 10:15 | SVZ | Depression | Phagocytosis |
| MG 1.9 18-042 | F | 55 | 6.83 | 09:15 | SVZ | Bipolar disorder | Phagocytosis |
| iPSC1 | M | 65 | N/A | N/A | iPSC, oMG, fibroblasts | Control | All experiments |
| iPSC2 | M | 66 | N/A | N/A | iPSC, fibroblasts | Control | RNAseq |
| iPSC3 | F | N/A | N/A | N/A | iPSC, oMG | Control | RNAseq, Stimulation, Phagocytosis |
| iPSC5 | M | 59 | N/A | N/A | Fibroblasts, oMG | Control | RNAseq, Stimulation, Phagocytosis |
| iPSC6 | F | N/A | N/A | N/A | iPSC; fibroblasts | Control | RNAseq |

*M* Male, *F* Female, *CSF* cerebrospinal fluid, *MFG* medial frontal gyrus, *STG* superior temporal gyrus, *THA* thalamus, *SVZ* subventricular zone, *iPSC* induced pluripotent stem cell, *MG* microglia donor, *MACS* magnetic-activated cell sorting

unsupervised hierarchical clustering, Spearman correlation coefficients, and principal component analyses (PCA) (Fig. 3a, b; Supplementary Fig. 4c) showed that both at day 38 and day 52 oMG closely resemble adult MG1. Further analysis revealed that oMG at both timepoints express well-established microglia markers (i.e., *AIF1, SPI1, ITGAM, RUNX1, TLR4, CSF1R, IRF8*) (Fig. 3c). We found that expression of part of these genes, like *AIF1* and *RUNX1*, is comparable between oMG and MG1, but for some the levels are threefold (*PTPRC* or *CX3CR1*) or even 10–100-fold different (*TREM2, P2RY12, TMEM119*) (Fig. 3d). To analyze whether microglia continued to mature within the organoid, we also analyzed the expression of these markers at 119 days in culture. Microglia from iPSC donors 1, 3, and 5 at day 119 vs. day 52 showed an increase in expression of all these selected typical microglia genes *AIF1, RUNX1, PTPRC, CX3CR1, TREM2, P2RY12,* and *TMEM119* (Fig. 3e).

The effect of the organoid CNS microenvironment on the development and differentiation of oMG was further evaluated by comparing the transcriptional landscape of these cells to previously published microglia datasets. For this purpose transcriptomic data from fetal microglia (fetal MG)[24], a 2-dimensional (2D) iPSC-microglia model (iPSC MG 1)[24], and data from other acutely isolated adult microglia (adult MG 2)[25] was included. The transcriptomes were compared for all available transcriptome data, but also using a panel of transcription factors that are expressed by microglia in vivo[26]. The study by Gosselin

et al.[26] describes that in the absence of other CNS cell types, microglia lose expression of these adult microglia-related transcription factor families. Unsupervised hierarchical clustering of all available transcriptome data revealed that both day 38 and day 52 oMG cluster with adult MG1 and previously published adult MG2, whereas iPSC MG clustered with fetal MG (Fig. 3f). Interestingly, iPSC MG and fetal MG first clustered with iPSC and fibroblasts prior to clustering with the oMG and adult MG 1 and 2 (Fig. 3f). Despite this clustering, Spearman correlation coefficients showed that the transcriptomic profiles of all microglia and microglia-like cells were relatively similar ($\rho = 0.7 \pm 0.07$) (Supplementary Fig. 4d). The Spearman correlation analyses of the transcriptomes of oMG day 52, adult MG1, iPSC MG and fetal MG by using microglia-specific transcription factors[26] (Supplementary Table 2) showed a strong correlation between iPSC MG and fetal MG ($\rho = 0.96$), whereas transcription factor profiles of oMG correlated most with those of adult MG1 ($\rho = 0.8$). The correlation between adult MG1 and fetal MG was low ($\rho = 0.37$) in this analysis (Supplementary Fig. 4e).

Taken together, transcriptome comparisons showed that oMG strongly correlate with adult MG1 and 2, whereas the 2D iPSC MG model highly correlate with fetal MG.

To further analyze differences in gene expression between oMG and adult MG, we analyzed transcriptomic differences between day oMG and adult MG1 (Supplementary Fig. 4f–h). Sample variability and size resulted in high Log2fold changes and

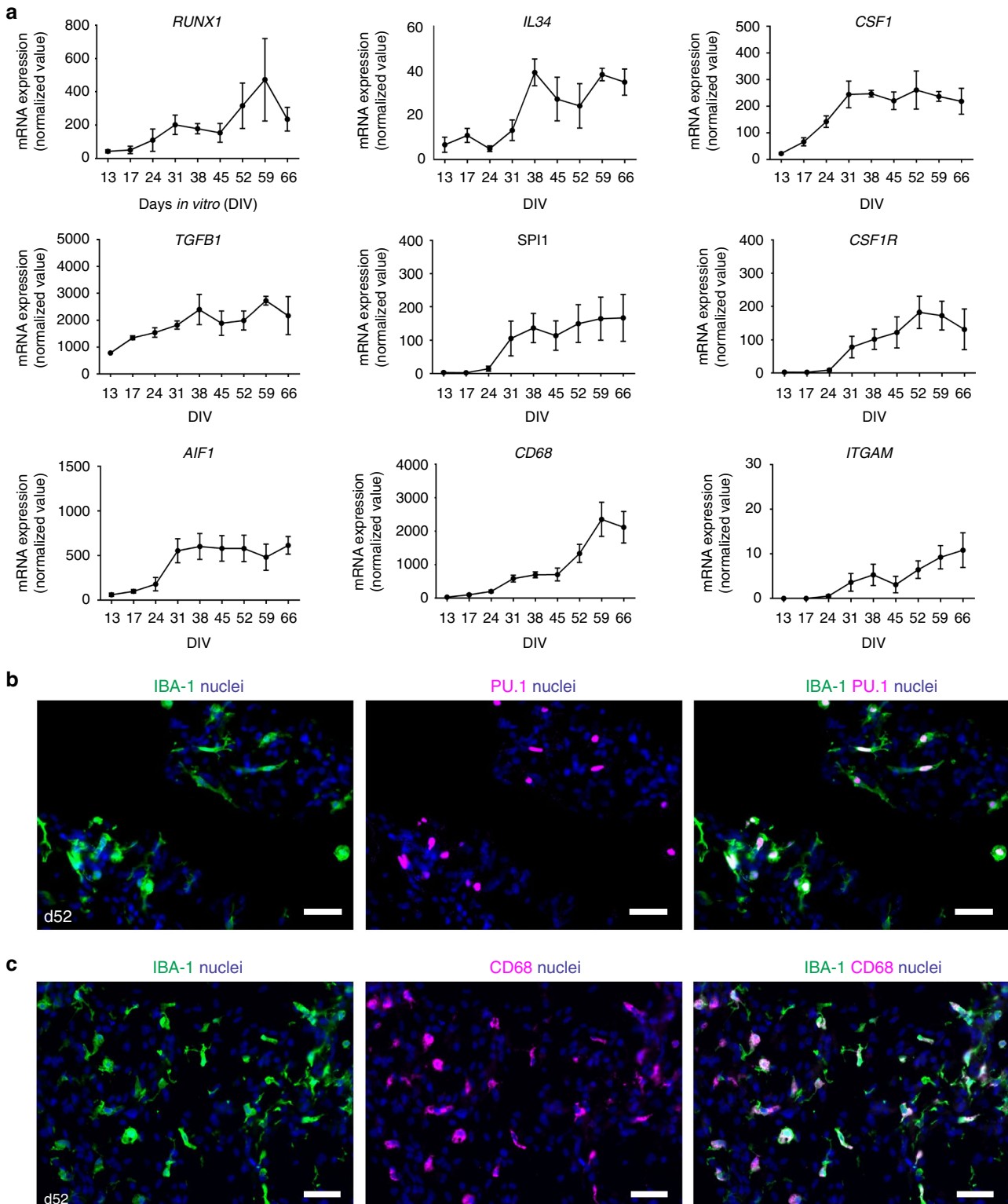

**Fig. 2** Microglia-specific gene and protein expression. **a** Time course of mRNA expression levels of early microglia markers and factors involved in microglia development in organoids assessed by qRT-PCR and normalized to the geomean of the reference genes *SDHA2* and *ACTB*. Data points represent the mean of four batches of two organoids per batch per timepoint. All batches consisted of organoids derived from iPSC 1. Error bars represent the standard error of the mean (SEM). See also Supplementary Fig. 3a. **b**, **c** Double immunostainings of IBA-1 combined with nuclear PU.1 (**b**) and CD68 (**c**) at day 52. Representative pictures of cerebral organoids from iPSC 1 are shown. Scale bars 40 μm

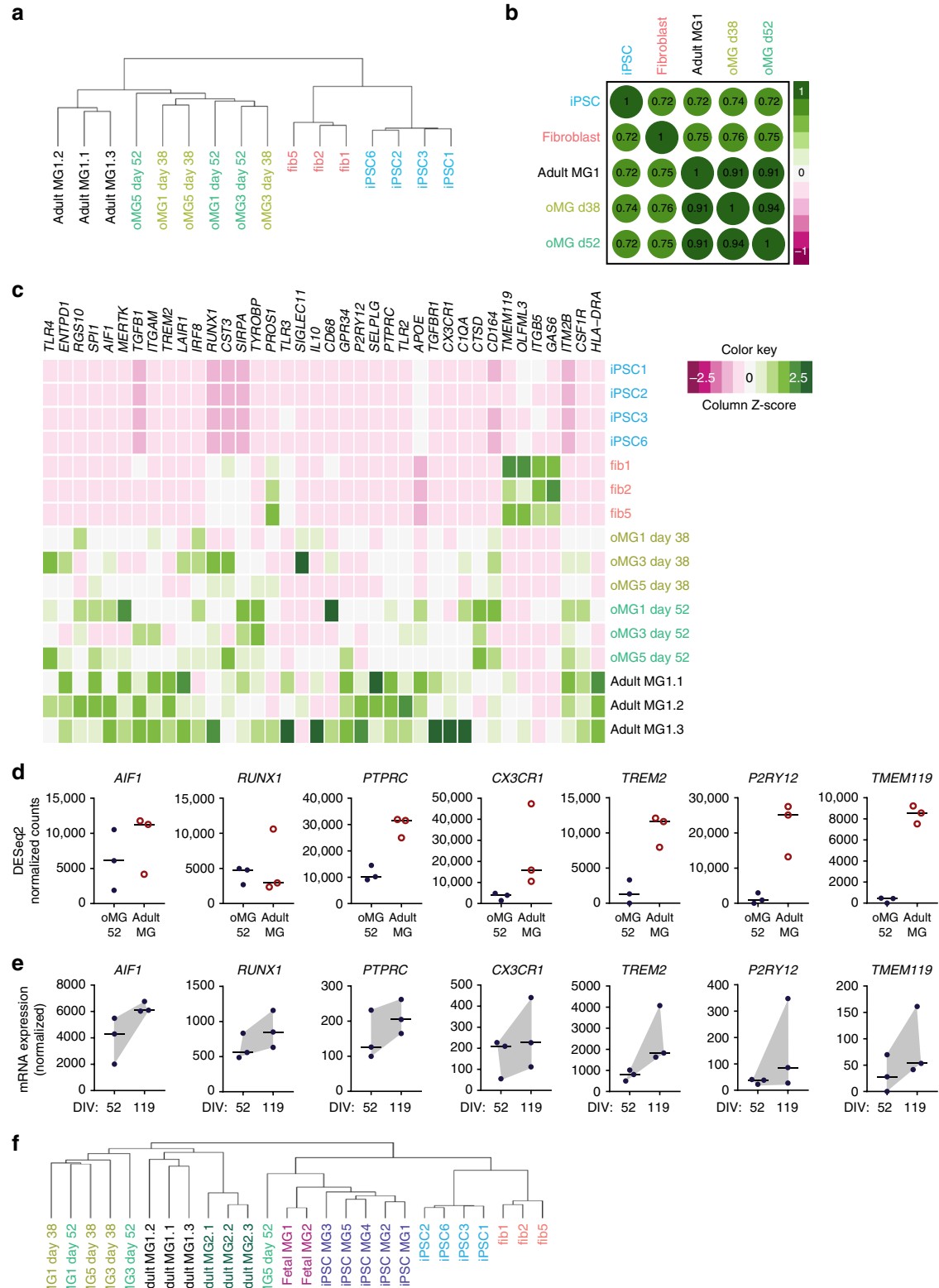

therefore shrinkage correction was performed. 1220 genes remained that were significantly upregulated in oMG day 52 and 1271 genes in MG1 (FDR < 0.05) whereas 2400 genes were significantly upregulated in oMG at day 38 and 2070 in adult MG1 (Supplementary Fig. 4i, j). Top 100 upregulated genes in oMG day 52 included *FN1, CDH4, COL1A2* and *COL3A1* for oMG and, *CDHR1, CXCR1* and *ACKR2* for adult MG1

(Supplementary Data 1). Functional gene classification with Panther GO-Slim[27] on the top 100 upregulated genes in both day 52 oMG and adult MG showed that the significantly upregulated genes in both samples belong to similar biological processes suggesting only subtle differences between oMG and adult MG. GO-pathway analysis on all DEGs did not show any significantly enriched pathways in adult MG1 vs. oMG, whereas oMG showed

**Fig. 3** oMG microglia gene expression profile is highly comparable to microglia(-like) cells. **a** Unsupervised hierarchical cluster analysis on DESeq2 rlog transformed raw counts of oMG day 38, oMG day 52, adult MG1, iPSC, and fibroblasts based on all genes after removal of common genes (FDR > 0.05, sum of raw read counts > 0) between samples. **b** Spearman correlation matrix for the correlations between oMG day 38, oMG day 52, adult MG1, iPSC and fibroblast DESeq2 rlog transformed raw counts of genes used in Fig. 3a. Median rlog gene counts of the biological replicates were used as input. The size and color of circles show the strength and direction of the correlation, respectively. **c** Heatplot representation of DESeq2 normalized expression levels for several microglia markers[23, 24, 34] in iPSCs fibroblasts, oMG day 38, oMG day 52, and adult MG1. **d** Plotted DESeq2 normalized expression of a selection of microglia typical genes in oMG at day 52 and adult MG1. **e** Temporal mRNA expression of characteristic microglia markers. Microglia were enriched with CD11b-MACs from organoids after 52 or 119 days in culture. mRNA levels were assessed by qRT-PCR and normalized to the geomean of the reference genes SDHA2 and ACTB. Data points represent mRNA levels of oMG 1, 3, and 5. **f** Unsupervised hierarchical cluster analysis on log transformed FPKM values for all available gene expression data of oMG day 38 and 52, adult MG1, fetal MG, another iPSC-derived microglia model (iPSC MG; GSE85839)[24] and additional primary adult microglia (adult MG2; GSE73721)[25]. Prior to hierarchical clustering, log transformed FPKM values were scaled for each sample. See also Supplementary Fig. 4, and Supplementary Data 1 and 2

enrichment for "collagen catabolic process", "extracellular matrix organization", "negative chemotaxis", "positive regulation of synapse assembly", and "endodermal cell differentiation" compared to adult MG1 (Supplementary Data 2). Myeloid function-related terms were detected by GO-pathway analysis of common genes (between adult MG1 and day 52 oMG), such as "nervous system development", "interleukin-12-mediated signaling pathway", and "neutrophil degranulation" (Supplementary Data 2).

Thus, both at the whole transcriptome level and when comparing microglia-specific factors and biological pathways, oMG are very similar to adult MG.

To investigate a potential developmental effect on gene expression between day 38 and day 52 oMG, differentially and commonly expressed genes and pathways were assessed. The transcriptomes of day 38 and day 52 oMG were most similar with 2160 differentially expressed genes (Supplementary Fig. 4k, FDR < 0.05). Due to this lower number, no significantly enriched pathways were detected when comparing day 38 and day 52 oMG. Several development-related terms were shared between day 38 and day 52 oMG, such as "nervous system development", "neutrophil degranulation", "cell adhesion", and "receptor internalization" (Supplementary Data 2). The top 100 significantly enriched genes in day 38 and day 52 oMG are depicted in Supplementary Data 1.

**oMG functionally resemble adult microglia.** For further characterization of oMG, single-cell suspensions were prepared from organoids at day 52 and compared to adult MG. The classical microglia cell surface markers CD45, CD11c, CD14, HLA-DR, CX3CR1, CD206 were all present in oMG and displayed an expression profile similar to that found in adult MG 1, as measured by flow cytometry (Fig. 4a).

For cell culture purposes, oMG were enriched using CD11b-coated beads and magnetic cell sorting (MACS) on whole organoid cell suspensions. MACS-sorted oMG and adult MG were cultured for > 7 days, and displayed a highly similar morphology (Fig. 4b). To assess if oMG and adult MG responded similar to inflammatory triggers, E. Coli-derived lipopolysaccharide (LPS; pro-inflammatory) and dexamethasone (anti-inflammatory) were added to the culture medium 24 h after plating the cells. Their immune response was evaluated by qRT-PCR after stimulation with LPS and dexamethasone for 6 and 72 h, respectively. Similar to adult MG, mRNA expression of *IL6* and *IL1B* was significantly increased in oMG (day 52) after exposure to LPS (Fig. 4c). This inflammatory response of oMG was significantly increased compared to adult MG (*IL6* and *IL1B*: $U = 0$, $n = 4$, $p < 0.05$). Treatment with dexamethasone induced the expected increase in mRNA expression of *CD163* and *MRC1* with a similar increase in fold change in oMG and adult MG (Fig. 4d). To test phagocytic capacity, we exposed oMG day 79 and adult MG to iC3b-coated beads for 0.5 and 1 h. Beads were detected in adult MG

and oMG and as expected the number of phagocytosed beads increased in time (Fig. 4e, f). This suggests that phagocytosis regulated via the C3 receptor is functional in oMG.

Thus, oMG and adult MG display functional inflammatory and phagocytic responses, in line with the similarity in their transcriptomes.

**oMG surround neurons and elicit immune response in organoids.** Microglia mediate inflammatory responses in the CNS, but are also known to directly affect neurons, and vice versa, in the developing human as well as rodent CNS[4,6,28,29]. Recently, it has been established that microglia can initiate synaptogenesis, but also eliminate synapses during development and in disease[5,8,29]. Here, we began to probe the applicability of cerebral organoids for studying the interplay between human microglia and neurons, and for studying microglia-mediated inflammatory responses in situ.

Immunohistochemistry for IBA-1 and TUJ1 followed by epifluorescent microscopy showed that although during early development (day 31) oMG were still in an amoeboid state these cells were already located in close proximity to neurites (TUJ1) (Fig. 5a). Upon further ramification, these oMG and their processes became more intertwined with neuronal processes (day 52, Fig. 5a). As microglia have been reported to regulate synaptic morphology and function we tested whether functional synapses are present in the organoids. Indeed, postsynaptic responses to presynaptic release of glutamate were detected at day 52 as evidenced by spontaneous excitatory postsynaptic currents (sEPSCs) (Supplementary Fig. 1q). Further, voltage-dependent sodium channels were present as inward currents were registered in response to evoked increases in membrane potential (Supplementary Fig. 1r, s). These data suggest that cerebral organoids contain functional synapses and could be used for studying the potential effect of microglia on synapse function. But can oMG processes be found in close proximity to neuronal synaptic structures? To address this question, we applied super-resolution stimulated emission depletion (STED) microscopy to study the distribution of microglia (IBA-1) in relation to the postsynaptic marker PSD-95 in cerebral organoids. High-resolution images from the STED experiment showed significant overlap between IBA-1 and PSD-95 signals, with PSD-95 material either inside IBA-1+ oMG processes or being in contact with or partially engulfed by oMG ramifications (Fig. 5b).

To further probe the potential use of the oMG-organoid model, we assessed whether or not the organoids responded to inflammatory compounds in situ by exposing complete organoids to LPS at day 52. We found that an acute cytokine response can be assessed 24 h after LPS stimulation (Fig. 5c). Release of IL6, TNF-α, but not IL10, was significantly increased after 24 h (IL6 and TNF-α: $W = 21$, $p = 0.03$) and 72 h (IL6 and TNF-α: $W = 21$, $p = 0.03$) of LPS stimulation, as measured by enzyme-linked immune assays

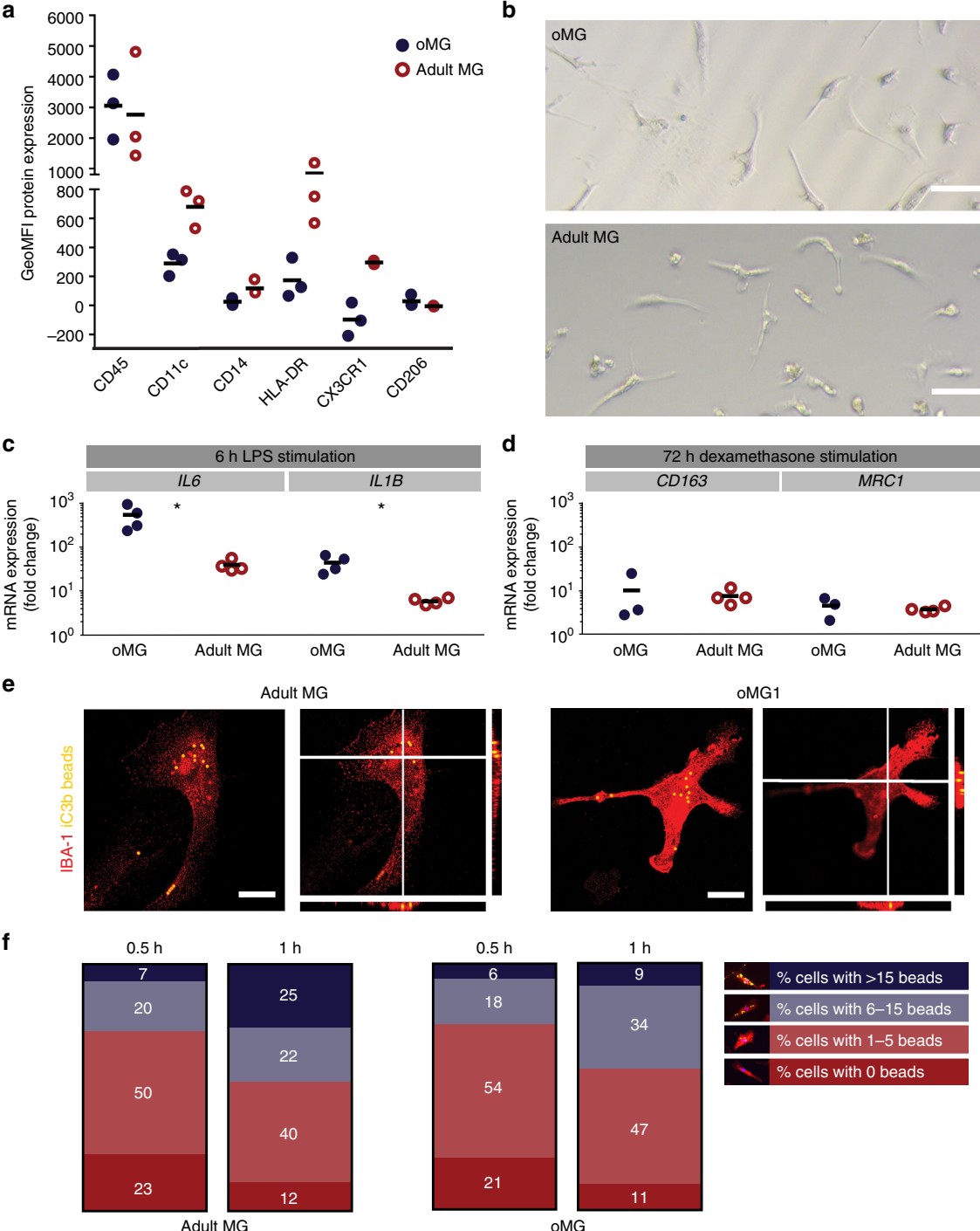

**Fig. 4** oMG expressed microglia-characteristic cell surface markers and showed similar functional immune and phagocytic properties as adult MG. **a** Flow cytometric analyses of the expression pattern of microglial extracellular markers on CD11b+-gated oMG (oMG 1, 3, and 5) compared to adult MG derived from three separate brain regions from adult MG1.1. (eight organoids were pooled per donor (oMG 1, 3, and 5) after 52 days in culture). **b** Morphology of magnetic automated cell sorted CD11b+ oMG 1 and adult MG in bright field microscope after 1 week in culture. Scale bar 40 μm. **c** mRNA expression, determined by qRT-PCR, of pro-inflammatory cytokines IL6 and IL1B after 6 h stimulation with LPS was significantly higher in oMG compared to adult MG (Mann–Whitney test IL6 and IL1B: $U = 0$, $n = 4$, $p = 0.03$). LPS-stimulated response relative to control condition without LPS. ($n = 4$ experiments, eight organoids pooled per experiment; adult MG1.1) (*$p < 0.05$). **d** Anti-inflammatory response of oMG and adult MG was compared by qRT-PCR for expression of anti-inflammatory genes CD163 and MRC1 upon 72 h stimulation with dexamethasone. Dexamethasone-stimulated response relative to control condition without dexamethasone. (oMG, $n = 3$ separate experiments in which oMG were isolated from > 4 pooled cerebral organoids from iPSC 1 per experiment; adult MG, $n = 4$). **e** Phagocytosis capacity was tested oMG 1 and adult MG by performing a phagocytosis assay with iC3b-coated green-yellow fluorescent beads. Phagocytosis was analyzed by confocal microscopy. Maximum intensity projection and orthogonal views are depicted. One experiment per donor. Scale bars 40 μm. **f** Quantification of iC3b beads engulfment by oMG and adult MG after 0.5 and 1 h exposure to the beads. Three randomly selected view fields per condition were manually quantified (oMG 1, for 0.5 and 1 h; adult MG1.7, 1.8, and 1.9, for 0.5 and 1 h). IBA-1+ cells were categorized based on the number of inoculated beads as follows: 0 (type 1), 1–5 (type 2), 6–15 (type 3), or > 15 (type 4)

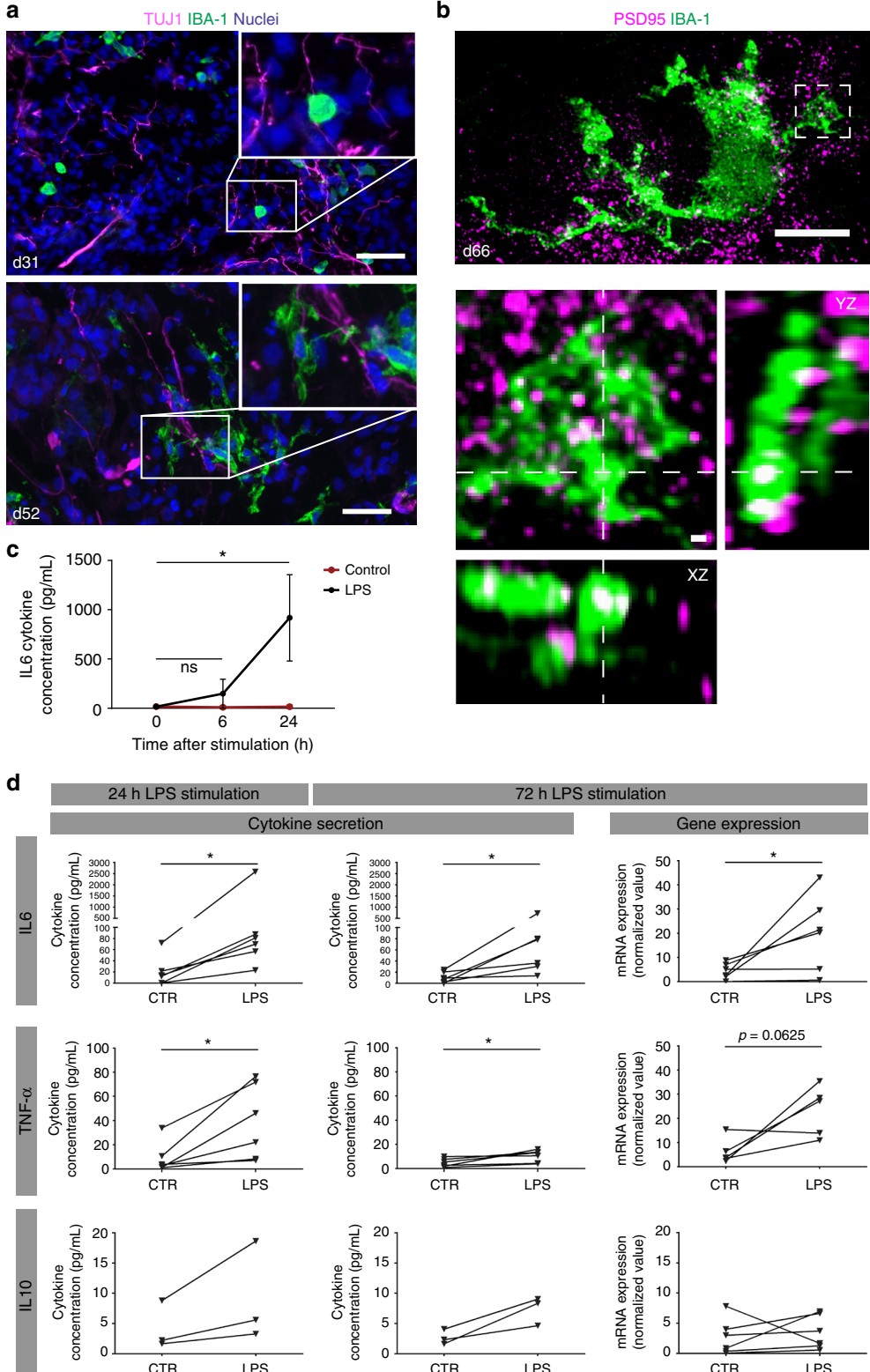

(ELISA) indicating an acute and long-term effect (Fig. 5d). Increased mRNA levels of *IL6* and *TNF*, but not of *IL10*, were detected after 72 h of LPS stimulation (IL6: $W = 21$, $p = 0.03$; TNF-α: $W = 19$, $p = 0.06$) (Fig. 5d). The cytokine response to LPS detected in the medium was low as compared to assays performed with isolated oMG. However, this is expected considering that only

~1% of all cells of an organoid are microglia (~10.000 cells) and that 100.000 cells were used in the 2D experiments.

Overall, these data support the idea that the organoid model described in this study represents a valuable tool to facilitate research into the interaction of different cell types in the human brain and the role of microglia during human CNS inflammation.

**Fig. 5** oMG form functional interactions with neurons and respond to inflammatory stimuli in situ. **a** Microglia (IBA-1)–neuron (TUJ1) interaction at an early (day 31) and late (day 52) timepoint was visualized by immunostainings. Representative pictures of cerebral organoids from iPSC 1 are shown. Scale bars 40 μm. **b** gSTED microscopy showing synaptic content inside microglial processes visualized by immunohistochemistry for IBA-1 and PSD-95 on day 66 organoids. Maximum intensity projection of the entire cell (scale bar 10 μm), and close-up of region of interest (box in dashed line) with maximum intensity projection (scale bar 1 μm) and orthogonal view (voxel size 0.18 μm). **c** Pilot experiment in which the IL6 inflammatory response was measured in organoids in situ. The cytokine response was significantly increased after 24 h as determined with a Friedman's ANOVA test ($p = 0.03$; Dunn's test $t = 6$ vs. baseline: -3 with $p = 0.44$; Dunn's test $t = 24$ vs. baseline: -6 with $p = 0.03$). $N = 3$ iPSC 1 organoids per group. Error bars indicate standard deviation. **d** Inflammatory response in situ was measured in whole organoids upon 24 and 72 h of LPS stimulation by ELISA for cytokine release (IL6, TNF-α, and IL10; both timepoints) and transcript fold change by qRT-PCR (after 72 h) for mRNA levels (IL6, TNF, IL10) in day 52 organoids. Secretion of both IL6 and TNF-α, but not IL10, was significantly increased after 24 and 72 h as analyzed with Wilcoxon matched-pairs signed rank test (IL6 and TNF-α: $W = 21$, $p = 0.03$). Similarly, mRNA levels of *IL6* and *TNF*, but not *IL10*, were increased after 72 h stimulation (*IL6*: $W = 21$, $p = 0.03$; *TNF*: $W = 19$, $p = 0.06$). mRNA levels were determined by qRT-PCR and normalized to the geomean of the reference genes *SDHA2* and *ACTB*. Two batches of cerebral organoids from iPSC 1, 3, and 5 were used ($n = 6$). IL10 levels were below detection level for the second batch of organoids. (*$p < 0.05$)

## Discussion

The current consensus is that cerebral organoids mainly consist of differentiated cells from the neuroectodermal lineage. This is a commonly mentioned drawback of this otherwise innovative model as cells derived from other lineages, such as microglia, play a crucial role in brain development, physiology, and pathology[11,14]. Our data refutes this assumption, as we show that microglia can innately develop within cerebral organoids. Single-cell RNAseq analyses of cerebral organoids by Quadrato et al.[18] did show the presence of mesodermal progenitors in cerebral organoids, but these did not display signs of microglial differentiation. In our study, instead of adding extra BDNF to the differentiation cocktail, like Quadrato et al., we reduced levels of the neuroectoderm stimulant heparin and delayed matrigel embedment of the organoids. Although we also detected oMG following the original protocol of Lancaster et al.[17], we found that this adaptation improved the yield of microglia. These oMG display a characteristic ramified morphology, express microglia-specific markers, mediate inflammatory responses, and have phagocytotic capacity.

The development of in vitro models to study human microglia has been a major challenge in the field. Primary microglia can be isolated from fresh human brain tissue[30,31] and microglia can be derived from iPSCs[24,32,33], but both methods have their own specific limitations. For example, acutely isolated microglia from post mortem brain or surgery tissue yields a limited number of cells. Another confounding factor is the quality of the post mortem tissue, and phenotypic changes induced by 2D in vitro culture[26]. Several recently published human iPSC-derived microglia differentiation protocols[24,32,33] partly circumvent these limitations. However, although these studies represent great advances in the field, they still do not provide for the 3D interaction of microglia throughout brain development with neurons and astrocytes in a human physiological setting. To generate microglia from iPSCs, protein cocktails are used that are known to drive microglia development in rodents, such as CSF1, IL34, and TGFβ1[1,2,34]. Interestingly, these factors are innately expressed by cerebral organoids as shown in the present study. It is thought that these previously identified factors act in concert with other yet undefined factors secreted or expressed by neurons and astrocytes to induce microglia differentiation and regulate microglia phenotype. This idea is supported by a recent paper describing that co-culture of iPSC-derived microglia with neurons changes the phenotype of microglia into a more homeostatic one[35]. Therefore, although every method to isolate or generate human microglia will have its own limitations, the advantage of oMG may be that these cells develop simultaneously with other cell types that provide for essential molecular factors in a 3D brain environment.

To validate the microglia identity of oMG, cells were phenotypically and functionally characterized and compared to adult MG. At a cellular level, oMG resembled adult MG in several ways. For example, when comparing the expression levels of extracellular surface markers, the response of oMG and adult MG to inflammatory stimuli, and the phagocytic response to iC3b-coated-particles. Functional synapses were detected in the organoids and microscopy-based approaches detected microglial ramifications in close proximity to neuronal processes and synaptic material. Different mechanisms of synaptic pruning have been described, including synaptic stripping, phagocytosis, and trogocytosis[29,36]. Our super-resolution microscopy data detected postsynaptic material inside and in close proximity to oMG, suggesting these different mechanisms may be at play as has been reported for MG. Future studies are needed to dissect these mechanisms for oMG in cerebral organoids, but our data further identify the organoid model as a valuable tool for studying human microglia-neuron interactions in 3D brain tissue.

At the transcriptome level, comparison between oMG and adult MG1, showed that oMG expressed key microglia markers (e.g., *SPI1*, *AIF1*, *CSF1R*, *TLR4*) at similar levels to MG found in human post mortem samples. Some key genes like *P2RY12*, *TREM2*, and *TMEM119* that were lowly expressed in oMG cultured for 52 days compared to adult MG, increased when oMG were held in culture longer (119 days). Whole transcriptome comparison with other datasets, including adult MG from this study, and previously published datasets of adult MG, fetal MG and iPSC MG, consistently validated the microglial identity of oMG. Our data replicated the finding of Muffat et al. that iPSC MG cluster with fetal MG rather than adult MG[24]. When comparing a selection of transcription factors that are important for microglia in vivo[26], adult MG had the lowest correlation with fetal MG, which could reflect a maturation effect. Simultaneously, oMG strongly correlated with adult MG, whereas fetal MG correlated with iPSC MG. Still, comparisons between multiple available datasets, including the ones in this study, have inherent limitations due to differences in sample processing of the different resources. Nevertheless, it is tempting to speculate that the mature phenotype of oMG may result from their interaction with other cell types in 3D, interactions during differentiation not present in recently published iPSC MG studies.

Our finding that microglia are present in cerebral organoids provides novel opportunities for studying the human brain. First, this model can be used to study the development of human microglia in association with other developing human brain cells. Gene editing of the iPSCs can be used to study the role of specific factors that mediate the interaction between neurons and glia. Second, this model will be useful to investigate how microglia are involved in the development of human neurons and astrocytes,

including processes such as neurogenesis, synaptogenesis, and synaptic pruning[3–5]. Third, this human cell model can be used to study the pathogenesis of brain infections by viruses such as Zika virus and HIV-1. Microglia are considered to be an important target for these viruses and their immune reaction is likely to play a role in the brain pathology caused by these pathogens[37]. Fourth, this model can be used to understand the role of microglia, macroglia and neurons in neurodevelopmental diseases, especially diseases with a clear genetic component, such as autism and schizophrenia. Fifth, it opens new avenues to study diseases such as Alzheimer's disease and ALS in which a role for microglia and their interaction with astrocytes and neurons is increasingly recognized[7].

Although the organoid model presented here is the first method to develop iPSC-derived human microglia in a 3D neuronal niche, it has a number of limitations[17]. An important limitation is the fact that organoids are still far from representing the human brain as they, for example, lack vasculature and cross-talk with other tissues in the developing embryo[17]. Another limitation is that the heterogeneity of cell types in cerebral organoids, due to the absence of SMAD inhibition, co-occurs with an increase in organoid-to-organoid variability. Although microglia where detected in all organoids generated in our study, their distribution was variable. A promising application to limit variability instigated by the use of multiple iPSC donors is the use of isogenic controls to study the role of specific proteins or genetic risk variants on microglia development, phenotype and function. In addition, although it is important to study microglia in the 3D context, there are inherent technical challenges as compared to studying these cells in 2D cultures. Fortunately, many tools have been developed in the past decade to study specific cell types in complete tissue, such as tissue clearing[38], cell-type specific expression of fluorescent reporters, and advanced microscopic techniques, including light sheet, super-resolution, and live-cell imaging[29]. Similar approaches can be employed to study microglia in the organoid in situ, but these will need further optimization.

In conclusion, here we report a method to develop iPSC-derived microglia in a 3D neuronal niche. The potential to study neuron/microglia interactions within human brain tissue will significantly further our knowledge of the structural, functional, and molecular mechanisms underlying microglia (dys)function in the intact and diseased human brain.

## Methods

**Human subjects**. The Medical Ethical Committee of the University Medical Center Utrecht granted approval for iPSC line generation. Subjects gave written informed consent. Fresh post mortem brain samples were provided by The Netherlands Brain Bank (www.brainbank.nl, Table 1). The autopsy was performed with signed informed consent (Table 1). Controls were individuals without diagnosis of a psychiatric or neurologic disorder.

**iPSC generation and characterization**. Whereas iPSC-line 1 and iPSC-line 3 have previously been described[19], iPSC-lines 4, 5, and 6 were similarly generated but have not yet been published. Briefly, skin fibroblasts were transfected with a lentiviral vector expressing C-MYC/SOX2/NANOG/KLF4 and seeded on mouse embryonic fibroblasts in hESC20 medium containing DMEM-F12 (ThermoFisher, 11320-074), 20% KOSR (ThermoFisher, 10828028), 1% NEAA (ThermoFisher, 11140-035), 1% L-Glutamine (ThermoFisher, 25030-024), 0.5% P/S (ThermoFisher, 15140-122), 496 μM ß-mercaptoethanol (Merck-Schuchardt, 805740), 20 ng/mL bFGF (ThermoFisher, AA10-155). Clones were characterized by immunocytochemistry with StemLight kit (Cell Signalling, 9656S) and quantitative reverse transcription PCR (qRT-PCR) for stem cell markers (relative to hUES6 line, Harvard University, RRID:CVCL_B194). Cells were karyotyped by G-band staining. Pluripotency was evaluated by embryoid body formation followed by 5 weeks of spontaneous differentiation in minimal medium DMEM-GlutaMax (Thermo-Fisher, 31966-021), 10% FBS (Sigma, F7524) and stained for markers of the three germ layers.

**iPSC culture**. Lines were maintained at 37 °C with 5% $CO_2$ in feeder free conditions. Cells were cultured on Geltrex-coated (Thermofisher, A1413202) dishes in mTeSR1 medium (Stem cell Technologies, 85850), weekly passaged with Accutase (Thermofisher, A11105-01) and seeded in mTeSR1 supplemented with ROCK-inhibitor Y-27632 (4.82 μM, Axon 1683). All lines were kept in culture up to 60 passages and frequently tested for mycoplasma infection (Lonza, LT07-318).

**Organoid differentiation**. Organoids were differentiated using a modified version of a previously published protocol (Supplementary Table 1)[12]. In short, after dissociation into single-cell suspension with Accutase, $3.5 \times 10^6$ cells were seeded per well in an Aggrewell800 microwell plate (StemCell Technologies, 27865) in 2 mL of hESC4 (4 ng/mL bFGF) medium supplemented with ROCK-inhibitor Y-27632 (48.2 μM). Approximately 11,500 cells per microwell were left to form embryoid bodies overnight. After 48 h, embryoid bodies (EB; 325 +/- 40 μm) were transferred to ultra-low attachment 96-well plates (Corning, 3474). Medium was replaced at day 4 by hESC0 (hESC without bFGF) and at day 6 by neural induction medium (NIM). At day 13, organoids (570 +/- 50 μm) were embedded in matrigel (Corning, 356234) and kept in organoid differentiation medium without retinoic acid (RA). Four days later they were transferred to spinning bioreactors on a magnetic platform (27.5 rpm; Pfeiffer, 183-001). No changes were made in the composition of the medium compared to the original protocol with exception of Heparin in the NIM where we used 0.1 μg/mL instead of 1 μg/mL. All organoid cultures were qualitatively selected according to the guidelines described previously[17]. Seven batches were made from iPSC 1, and three batches of iPSC 3 and 5 (Table 1). Each batch had a minimum of 32 organoids.

**Single-cell preparation from organoids**. Five organoids per 60 mm dish were washed three times with DPBS (ThermoFisher, 14190094) and immersed in 6 mL of papain (18.6 U/mL, Worthington, LK003176) and DNAse 1 (337 U/mL Worthington, LK003170) in DMEM/F12. Mechanical dissociation using scalpels was followed by a 30 min incubation at 37 °C on a shaker. In all, 2% FBS was added to stop the enzymatic reaction and the resulting single-cell suspension was centrifuged followed by an additional incubation with DNAse 1 (337 U/mL, Roche Diagnostics, 11284932001) in MACs buffer (PBS pH 7.4 (Gibco Life technologies, MA)), 2 mM EDTA (Sigma-Aldrich, The Netherlands) and 1% FBS at room temperature for 15 min.

**Single-cell preparation from post mortem brain tissue**. Dissociation of fresh post mortem brain tissue started within 2 to 24 h after autopsy according to the protocol described before with a minor modification[30,39]. In short, single-cell suspension was achieved by mechanical dissociation followed by enzymatic digestion using either 200 μg/mL DNAse 1 (Roche Diagnostics, 11284932001) and 3700 U/mL collagenase type 1 (Worthington, LS004196) for the medial frontal gyrus, superior temporal gyrus and thalamus, or DNAse 1[30,40] and Trypsin for the subventricular zone[39]. The protocol was slightly adjusted by adding an extra DNAse 1 incubation at room temperature (RT) for 15 min after the first enzymatic digestion to improve the single cellularity before FACS-sorting.

**Fluorescence-activated cell sorting for RNAseq**. The single-cell suspension was incubated with an FC receptor blocking reagent (Miltenyi Biotec, 5170126102) (1:20), anti-CD11b (eBioscience, 12-0118-41) (1:80) and anti-CD45 (eBioscience, 11-9459) antibodies (1:20) at 4 °C for 15 min. Cells were washed twice and suspended in glucose-potassium-sodium buffer (GKN-BSA; 8 g/L NaCl, 0.4 g/L KCl, 1.77 g/L $Na_2HPO_4.2H_2O$, 0.69 g/L $NaH_2$ $PO_4.H_2O$, 6 g/L D-(1)-glucose, pH 7.4) with 0.3% BSA. 7-AAD (BD Biosciences, 5168981E) (1 μg/mL) was added for cell death detection. Cells were poured over a 70 μm cell strainer before sorting. Cells were sorted/gated that were alive, single, and CD11b$^+$/CD45$^+$ with the FACSaria III (Supplementary Fig. 5a). The sorted CD11b$^+$/CD45$^+$ fraction was placed on ice, centrifuged at 400 rcf for 5 min, lysed in Trizol reagent (Life Technologies, 15596018). The mean percentage of CD11b$^+$/CD45$^+$ cells of iPSC donors 1, 3, and 5 was 0.83% ± 0.3 at both timepoints for oMG (Supplementary Fig. 2c). RNA was isolated with the miRNeasy mini kit (Qiagen, 217004) according to the manufacturer's protocol including DNAse 1. Isolated RNA was eluded in 40 μL RNAse free water and concentration determined using VarioSkan Flash microplate reader (Thermo Scientific, MA).

**RNA sequencing**. Quality control of RNA samples, library preparation and sequencing of the oMG and adult MG1 samples was performed by the Service XS sequencing facility (Genome Analysis Facility, Leiden, The Netherlands). Single-end polyA enriched (adult MG1, oMG day 38 and oMG day 52) and paired-end rRNA depleted (fibroblasts, iPSCs) libraries were sequenced on the Illumina NextSeq 500 sequencer and on the Illumina HiSeq 2500 platform (Illumina, San Diego, California), respectively. Reads were de-multiplexed and converted to FASTQ files using bcl2fastq (version 2.17; adult MG1, oMG day 38 and oMG day 52) or CASAVA (fibroblasts, iPSC) software from Illumina. FASTQ files were mapped to the hg19/GCh37 reference human genome (iGenomes). For adult MG1, oMG day 38 and oMG day 52 Bowtie2 (version 2.1.0) and TopHat2 (version 2.0.14) software was used for read alignment. TopHat2 (version 2.0.13) was used for alignment of fibroblast and iPSC reads. Read counts per transcript were determined

using HTSeq software (version 0.6.1). RPKM/FPKM (reads/fragments per kilobase of exon per million reads mapped) values were calculated from raw read counts using library size and transcript lengths.

**Differential expression and gene ontology analyses**. Read counts per transcript for adult MG1 ($n = 3$), oMG day 38 ($n = 3$), oMG day 52 ($n = 3$), fibroblasts ($n = 3$), and iPSC samples ($n = 4$) were analyzed within the R environment (version 3.4.3) using DESeq2 (version 1.18.1). After normalization using DESeq2 default settings that correct for library size, differential gene expression was calculated between samples to identify common genes among samples (False discovery rate (FDR) > 0.05, sum of raw read counts > 0). Differential gene expression between oMG day 52 and adult MG1, and oMG day 52 and oMG day 38 was calculated by performing a pair-wise comparison using DESeq2. The analysis was corrected for sex. Significantly differentially expressed genes (FDR < 0.05) and common genes (FDR > 0.05, sum of raw read counts > 0) were identified for each comparison. Shrinkage of log2Fold changes was applied using the apeglm algorithm in DESeq2. To determine microglia-specific common and differentially expressed genes between oMG day 52 and oMG day 38, and oMG day 52 and adult MG1 the common genes between microglia, fibroblasts and iPSC were removed. Gene ontology analysis was performed to determine overrepresentation of the different gene sets in GO cluster biological functions (BP), using the topGO R package (version 2.30.1, Gene Ontology Consortium. Gene Ontology Consortium: going forward. Nucleic Acids Res. 2015; 43: D1049–56. doi: 10.1093/nar/gku1179). Fisher statistical test was performed to test for overrepresentation, with a weighted algorithm to correct for dependency between parent–child relations between gene ontology clusters. The test was performed in the context of all Ensembl IDs considered in the differential gene expression analysis. Annotation build of 12-July-2018 (org.Hs.eg.db, ID = ensemble) was used for the annotation. p-values were adjusted for multiple comparisons using the Bonferonni correction. Principal component analysis, scaling of expression values and unsupervised hierarchical cluster analysis were performed in R (version 3.4.3) and plots were generated using the ggplot2 package (version 3.0.0). For spearman correlation analysis, the Harrell Miscellaneous (Hmisc, version 4.1-1) package was used and plots were generated using the corrplot package (version 0.84) in R.

**Gene expression analysis with real-time PCR**. RNA was isolated and cDNA prepared as described before[30]. Primers were intron-spanning and designed with PrimerBLAST (NCBI). Absolute levels of expression were determined ($2^{\Delta CT}$) and normalized with the geomean of two reference genes *ACTB* and *SDHA2* (see primer sequences in Supplementary Table 3). Two organoids were pooled per timepoint for RNA isolation.

**Magnetic cell sorting (MACs) of oMG and adult MG1**. The CD11b$^+$ fraction of organoids was enriched by MACs according to manufacturer's protocol (Miltenyi, 130049601). In short, single cells were incubated with the beads in MACS buffer and sorted using magnetic columns. The CD11b$^+$ cells were collected in microglia culture medium (RPMI 1640 (Life technologies, 21875034), 10% FBS, 2 mM L-glutamine, 100 U/mL penicillin, 100 µg/mL streptomycin (BioWhittaker, Belgium), and 100 ng/mL recombinant IL34 (Miltenyi, 130105780)). Eight organoids were used per digestion to obtain sufficient yield for experiments at day 52, 74, and 119. On average ($n = 9$) the CD11b$^+$ cells constituted 5% +/- 2.8 of the entire single-cell suspension. Sorting efficiency was confirmed by assessing the expression of specific microglia- and non-microglia-related genes in three batches of oMG 1 and adult MG1 (Supplementary Fig. 3b). We detected *OLIG2* in the microglia-enriched fraction indicating the presence of a low percentage of oligodendrocytes that are known to express low levels of CD11b. The CD11b$^+$ MACs enriched oMG or pMG cells were plated 1*10$^5$ cells per well in a 96-well flat-bottom plate or 2*10$^5$ on a glass coverslip in a 24-well flat-bottom plate and cultured in microglia-culture medium at 37 °C in 5% CO$_2$,

**Pro- and anti-inflammatory stimulation**. After one day in culture, adult MG and oMG were exposed to 100 ng/mL lipopolysaccharide (LPS) from *Escherichia coli* (Sigma-Aldrich, 0111:B4) or 1 µM dexamethasone (Sigma-Aldrich, D4902-25) for 6 h or 72 h, respectively ($n = 4$). Cells were lysed in 500 µL Trizol reagent.

**Preparation of iC3b-coated beads**. Human iC3b (Merck Millipore, 204863) protein was coated on FluoSphere® (ThermoFisher, F8827) Carboxylate-Modified Microspheres (2.0 µm, yellow-green fluorescent (505/515), 2% solids). Twenty-five microliters of bead suspension was washed twice by centrifugation (400 rcf, 10 min) in MES buffer (0.488 g MES hydrate (CAS # 4432-31-9 (anhydrous), Sigma-Aldrich, Missouri) in 50 mL MQ (50 mM), pH 6.0). 50 µL iC3b (50 µg protein), or 50 µL MES buffer (for the negative control) was added to the beads and incubated on a shaker at RT for 15 min. In all, 20 µL 10 g/L freshly prepared EDAC buffer (Thermofisher Scientific, MA) was added to the mixture for 2 h on a shaker at RT. The reaction was quenched with 100 mM glycine (100590, Merck Millipore, Germany) at RT for 30 min. Beads were washed three times in PBS by centrifugation (14,000 rpm, 10 min), suspended in 100 µL PBS and stored at 4 °C. The concentration was measured with a photospectrometer (BioPhotometer,

Eppendorf, Germany). Coating was confirmed by comparing phagocytosis of non-coated and coated beads.

**Phagocytosis assay**. After one week, cells, cultured on coverslips in a 24-well plate, were incubated with iC3b-coated beads at a concentration of 3 beads per cell at 37 °C for 0.5 or 1 h. Cells were washed three times with cold PBS, fixed with 4% PFA for 10 min at RT, and immunostained for IBA-1. Imaging was performed with an Olympus (Tokyo, Japan) Fluoview FV1000 confocal microscope or with a Zeiss Axio- Scope A1. Bead engulfment by adult MG and oMG was analyzed on three randomly selected view fields per section and per condition (oMG1, for 0.5 and 1 h; adult MG1.7, 1.8, and 1.9, for 0.5 and 1 h). IBA-1$^+$ cells were categorized based on the number of inoculated beads as follows: 0 (type 1), 1–5 (type 2), 6–15 (type 3), or > 15 (type 4) using FIJI version 1.49 software.

**Organoid fixation and immunohistochemistry**. Organoids were fixed in 4% PFA at 4 °C for 10 min, washed in PBS and transferred to 30% sucrose solution overnight at 4 °C. Samples were embedded in tissue-tek (CellPath, KMA0100-00A) snap-frozen on dry ice and stored at –80 °C. Twenty micrometersections were obtained with a cryostat.

Sections were blocked for 1 h in blocking solution (1% Triton-X, 3% BSA and 10% donkey or goat serum) at RT followed by incubation with primary antibodies in blocking solution overnight at 4 °C. Sections were then washed with PBS and incubated with secondary antibodies and Hoechst (ThermoFisher, H3569) at RT for 2 h. When Hoechst staining was not performed, samples were incubated with DAPI for 5 min before mounting. See Supplementary Table 4 for antibodies used in this study. Samples were mounted on glass coverslips using Fluorosave (CalBioChem, 345789) and imaged (Olympus (Tokyo, Japan) Fluoview FV1000 confocal microscope or Zeiss Axio- Scope A1. In all, ≥ 3 organoids from three batches were sectioned and imaged.

**Super-resolution STED microscopy**. Dual-color gated STED (gSTED) imaging of day 66 organoids immunostained for IBA-1 and PSD-95 was performed with a Leica TCS SP8 STED 3X microscope using a HC PL APO 100 × / 1.4 oil immersion STED WHITE objective. The 590 and 647 nm wavelengths of pulsed white laser (80 MHz) were used to excite the Alexa594-labeled PSD-95 and the Atto647N-labeled IBA-1, respectively. Both Alexa594 and Atto647N were depleted with the 775 nm pulsed depletion laser (10%–20% of maximum power) and we used an internal Leica HyD hybrid detector (set at 100% gain) with a time gate of $0.3 \leq tg \leq 6$ ns. Multiple Z-stack were obtained at 0.18 µm interval to acquire 4.0 µm image stacks (corrected for refractive index mismatch between oil immersion objective and FluorSave, the mounting medium used) in 2D STED mode using the 100x objective with ~2x zoom to obtain approximately 80 nm pixel size. Raw gSTED images collected using Leica TCS SP8 were subjected to deconvolution as stacks using Huygens deconvolution software. Deconvolution of the 594 and 647 channels was performed separately using the CMLE deconvolution algorithm, with a maximum of 40 iterations and the signal-to-noise ratio (SNR) set at 7.

**Quantification of immunohistochemistry**. Microglia perimeter was quantified using a macro in FIJI software on sections of four separate organoids per timepoint. The macro consisted of the following steps: transformation to 8-bit; scale set to µm; automated default threshold applied; converted to mask (particles filtered with size > 40 µm; option "outline"; the outline area was summarized and averaged over total number of microglia in the image to obtain the perimeter/microglia cell (Fig. 1f).

To determine the fraction of TUJ1 and IBA-1 staining in organoid sections, two tiled scans per organoid of three batches from oMG1 were prepared. A macro in FIJI was created with the following steps: background subtraction (rollingbal set to 10 (hoechst) or 78 (TUJ1, IBA-1) pixels; contrast was automatically enhanced; automated default threshold applied; particles were quantified with size of > 5 (hoechst) or > 10 (TUJ1, IBA-1) pixels; positive area was summarized; TUJ1 and IBA-1 was normalized by dividing the positive area by the hoechst positive nuclear area (Supplementary Fig. 2a, b, d).

**Pro-inflammatory stimulation cerebral organoids**. Organoids were singled in a 24-well flat-bottom plate. After 24 and 48 h, LPS (100 ng/mL) was added to the medium. A pilot experiment to determine the optimal timing for cytokine measurements, showed that 24 h after LPS stimulation the cytokine response was significantly increased as determined with a Friedman's ANOVA test ($p = 0.03$; Dunn's test $t = 6$ versus baseline: -3 with $p = 0.44$; Dunn's test $t = 24$ vs. baseline: -6 with $p = 0.03$) (Fig. 5c). Therefore, we used 24 h to measure the acute response, and 72 h for the effect on the long-term. Medium was collected after 24 h and 72 h for ELISA. Organoids were lysed in Trizol. Two batches of organoids from oMG 1, 3, and 5 were used for this experiment.

**Cytokine secretion**. Anti-human IL6, TNF-α, and IL10 ELISA Ready-Set-Go® kits (eBioscience, 88706688, 88734688 and 88734688, respectively) were used according to manufacturer's protocol. Samples were analyzed in a Varioskan™ Flash (ThermoFisher Scientific, MA) with an optical density (OD) of 450 nm. Blank

subtraction was performed and the value of the samples was based on the standard curve.

**Primer panels for microglia characterization**. A panel of microglia markers was selected based on literature[34,41,21–23] (Supplementary Table 3). In addition, general markers for astrocytes, neurons, and oligodendrocytes were selected to determine enrichment of the microglia population after CD11b+ sorting.

**Protein expression analysis**. In all, $8*10^5$ single cells from fresh human brain material and cerebral organoids were stained in a 96-well V-bottom plate using the following antibodies from Ebioscience (San Diego, CA): CD45 (#119459), CD11b (#12011841), CD14 (clone 61D3, #90170149025), CD11c (clone 3.9, #11011641), HLA-DR (clone LN3, #17995641), CX3CR1 (clone eB149/10H5, #17609941), IgG1 (#114714 and #17471441), IgG2b (#17403181); antibodies from BD Bioscience: CD206 (#561763); antibodies from BioLegend IgG2b (#400319). Protein expression was quantified on a FACSCanto II (BD Biosciences) and analyzed with FACSDiva software (Version 8.0.1, BD Biosciences). Protein expression was measured in cells that were gated to be alive, single, and CD11b+ (Supplementary Fig. 5b). The geoMFI mean was subtracted from the geoMFI of the corresponding isotype control.

**Electrophysiology**. Organoids were cut in two and continuously perfused with artificial cerebrospinal fluid at 32 °C ± 1, containing 120 mM NaCl, 3.5 mM KCl, 1.3 mM $MgSO_4$, 1.25 mM $NaH_2PO_4$, 2.5 mM $CaCl_2$, 10 mM D-glucose, and 25 mM $NaHCO_3$, and 5% $CO_2$, pH 7.4. Patch pipettes for recording were produced from borosilicate glass (1.5 mm outer diameter, 0.86 mm inner diameter; Harvard Apparatus Limited, Holliston, Massachusetts, USA; pipette resistance ~4–5 MΩ) on a P-97 Flaming/Brown micropipette puller (Sutter Instruments, USA) and filled with pipette solution containing (in mM): 120 Cs methane sulfonate, 17.5 CsCl, 10 Hepes, 2 MgATP, 0.1 NaGTP, 5 BAPTA; pH was 7.4, adjusted with CsOH. Whole cell voltage clamp recordings were performed using an Axopatch 200B (Molecular Devices, Sunnyvale, California, USA) amplifier. Responses were filtered at 5 kHz and digitized at 10 kHz using Digidata 1322 A (Axon Instruments, USA). Data was analyzed using pClamp 9.0 and Clampfit 9.2 (Axon Instruments). Recordings with a series resistance of less than 2.5 times the pipette resistance were accepted for analysis. The holding potential for spontaneous excitatory postsynaptic currents was kept at –65 mV. The voltage-dependent sodium currents were activated with depolarizing voltage steps with duration of 100 ms starting from –60 mV with increments of 10 mV to + 40 mV (Supplementary Fig. S2E). Series resistance was compensated for approximately 70%. Due to the very small voltage-dependent potassium and calcium currents, it was not necessary to block these currents.

**Statistics**. Statistical comparisons were performed with Friedman's ANOVA, two-tailed Mann–Whitney (non-paired data) and Wilcoxon matched-pairs signed rank (paired data) tests by using GraphPad Prism software version 6 (Graphpad Software, CA), as normality assumptions were not met. R version 3.0.2 software (CRAN:https://www.r-project.org) was used for RNA sequencing data analysis. Immunofluorescent images were analyzed and quantified by using FIJI version 1.49 software (NIH, Bethesda, MD). A significance level of $p < 0.05$ was used.

## Data availability

All relevant data is available from the authors. RNAseq data is submitted to the NCBI database with the record number GSE102335.

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

## Acknowledgements

We are grateful to the Netherlands Brain Bank (http://www.brainbank.nl) who provided us with the post mortem human brain tissue. We thank the MIND facility at the UMC Utrecht for their help. We are thankful to M.H.M. de Goeij for setting up the phago-cytosis assay and Dr. R. F. Hevner (University of Washington, Seattle, WA) for providing us with the TBR1 antibody. Furthermore, we want to thank Vanessa Marques Donega for providing us with adult MG for an experiment and Yujie He for helping with ELISA assays. Lastly, we are grateful for the advice on bioinformatics from Mark K. Bakker and Onur Basak. This project was financially supported by: The Netherlands Organization for Scientific Research (NWO-VICI; to E.M.H. and R.J.P.), Parkinson fund #937409 and ZonMW memorabel #733050107 (to E.M.H.); Utrecht University Strategic Theme Dynamics of Youth, Stichting ALS Nederland; Prinses Beatrix Spierfonds (to R.J.P); L.D. W: Narsad Young investigator grant.

## Author contributions

P.R.O. and R.V.S contributed equally in the project design, experiments, data collection and analyses. L.D.W., E.M.H., and R.J.P. contributed equally in project supervision and conceptualization. L.D.W., P.R.O., and R.V.S. wrote the manuscript. E.M.H. and R.J.P. provided critical feedback in reviewing and editing. All authors reviewed and gave input for the final version of the manuscript. E.J.B. performed all RNAseq analysis. H.K. performed electrophysiology experiments. M.A.M.S. and R.E.D. provided post mortem adult MG material and helped with setting up FACS of oMG. L.E.J., S.K., and E.R.M. helped with fibroblasts and iPSC culture and characterization. O.H. performed RNA isolation and sample preparation for RNAseq of iPSC and fibroblasts. A.B.B. helped with the organoid culture maintenance. N.S. and H.D.M. performed, data processed, and interpreted STED imaging. Funding for the project was obtained by R.S.K., L.H.B., E.M. H., L.D.W., and R.J.P.

## Additional information

**Competing interests:** L.H.B. declares the following competing financial interests: travel grants and consultancy fees at Baxter International, and is a member of the scientific advisory board at Biogen Idec, Cytokinetics, and no other competing interests. All other authors declare no competing interests.

