## [Peer Review File · Nature Communications]

Reviewers' comments:

Reviewer #1 (Remarks to the Author):

The manuscript by Ormel et al. provide compelling evidence that mature microglia-like cells are generated within cerebral organoids (oMG) and investigate the transcriptional expression profile and the function of oMG. Perhaps most interesting is that oMG closely mimic the transcriptome of adult microglia isolated from postmortem brain tissue, unlike iPSCs-derived microglia published recently by several groups. A 3D neural organoid model in which microglia are present along with other neural lineages would be essential for studying not only microglia development under physiological-like microenvironment, but also may have applications in studying neuron-glia interactions in health and disease. Therefore, this study would represent an important first step to achieve such model and an important addition to the field, provided several issues are addressed, as outlined below.

Although the data of microglia differentiation and efficiency do look impressive, my major concerns are that the MS somewhat suffers from lack of quantifications, fail to describe the overall properties of the cerebral organoids that have high/low efficiencies of generating microglia, as well as the precise description of the protocol that was used to achieve such seemingly highly microglia number within the organoids. The author mentioned briefly in the discussion section that minor modifications were applied to protocol published previously (Lancaster et al 2014), namely Matrigel embedding timing, and microglia numbers were higher under these conditions. However, no direct comparison or additional details were provided in the MS. I think this would be an important point to address since that this is the first report to describe this. In addition, given the heterogeneity of cerebral organoid (which is totally fine and not necessary represent disadvantage of the system over other published dual-SMAD based systems) it is not clear whether organoid that harbor cortical-like cytoarchitecture or more enriched with neural epithelium tissue also have similar microglia generation efficiency. From what I can tell, both the organoid that expresses/enriched with T gene and microglia markers do not display well-developed neuroepithelium structure that should be presented in optimal organoids at the studies organoids stages.

More specifically:

1. What is the ratio the organoids display microglia-like cells within the same batch and across different batches? Is there variability in the of percentage/number of Iba1+ cells between different organoids? Is there any correlation between batches display more non-neural tissue and microglia differentiation? It would be useful to provide a tile scan of whole sections of organoids stained for neural markers along with Iba1 to show the distribution and to demonstrate that these organoid are indeed neural organoids and not simply neurospheres.
2. What are the modifications from the Lancaster protocol if any? would be good to list those out in the text.
3. Figure 3: What genes are driving the differences between the clustering? What's the difference between adult MG and oMG?
4. It would be important to provide some kind of quantification of the levels phagocytosis following exposure of oMG to iC3b-coated beads in Fig 4.
5. In Fig. 6, LPS stimulation was applied for 24h, relative to 6h in oMG cultured in vitro. Does the inflammation response occur mainly after 24h in this experiment? Or similar induction can be achieved after 6h? Also, the majority of experiments showed low levels of cytokines induction. This can be explained by different levels of oMG within the organoids or alternatively low reactivity. This point can be addressed by correlation of the secretion/expression levels of cytokines with the average number of microglia cells in the organoids to clarify this point.
6. The upregulation of adult microglia transcriptome signature in oMG is impressive, however, from what I can tell, oMG lack/has a very low expression of key markers of adult microglia, for example, TMEM119, P2RY12, and TREM2. It would be important though to discuss this point. One would expect that prolonged culture of organoids beyond day 52 may achieve improve maturity. I am not necessarily suggesting analysis of transcriptome beyond day 52, but this point could be

addressed by RT-PCR/immunostaining for these markers.

7. The similarity between the oMG and adult MG in Fig3 is impressive, however, the difference between fetal and adult MG is only 0.37 which is quite surprising. Such difference can be partially explained by the fact that datasets were obtained from different resources and also can depend on the methods that the data were normalized, it would be important to explain and discuss this point.

Minor point:

1. Line 100, reference 12 should be reference 17 to refer to the original study.

Reviewer #2 (Remarks to the Author):

This manuscript by Ormel et al. describes a new cerebral organoid model in which microglial cells are found. These cells have a ramified morphology, a transcriptomic profile resembling mature microglia isolated from human postmortem brain tissue, and may also engulf synaptic materials as observed during normal physiological conditions. To my knowledge this is the first report of microglia spontaneously developing within cerebral organoids, which makes it novel and interesting to a broad readership.

Considering my expertise I will comment on the morphological aspects of this study, for which I have a few concerns as detailed below. In particular, without methodological information it is very difficult to assess whether the analyses of microglial morphology, phagocytosis, and synaptic pruning were properly conducted. It would be useful in the Suppl. materials to provide information pertaining to the imaging and analysis itself. Regarding the analysis of synaptic pruning, the complete engulfment of PSD-95 by IBA1-positive cells is not sufficiently convincing in my opinion, considering that confocal microscopy does not have the resolution required to discriminate between partial versus complete microglial engulfments. These findings should be strengthened. In particular, in Fig. 5b, the PSD-95 puncta appears to be encircled rather than engulfed. Recently, different mechanisms underlying pruning have emerged: synaptic stripping and phagocytosis, but also trogocytosis and extracellular digestion. Synaptic stripping is the physical separation of pre- and post-synaptic elements by microglial processes. Phagocytosis is their complete internalisation within endosomes fusing with lysosomes. Trogocytosis involves the removal of small pieces of presynaptic boutons rather than their engulfment as a whole, while extracellular digestion refers to the degradation of materials inside the extracellular space. Additionally, it would be useful to assess whether microglia in the organoids also interact with presynaptic boutons, in addition to dendritic structures.

Point-by-point response to reviewers' comments

Reviewer #1

The manuscript by Ormel et al. provide compelling evidence that mature microglia-like cells are generated within cerebral organoids (oMG) and investigate the transcriptional expression profile and the function of oMG. Perhaps most interesting is that oMG closely mimic the transcriptome of adult microglia isolated from postmortem brain tissue, unlike iPSCs-derived microglia published recently by several groups. A 3D neural organoid model in which microglia are present along with other neural lineages would be essential for studying not only microglia development under physiological-like microenvironment, but also may have applications in studying neuron-glia interactions in health and disease. Therefore, this study would represent an important first step to achieve such model and an important addition to the field, provided several issues are addressed, as outlined below.

We thank the reviewer for this positive assessment.

Although the data of microglia differentiation and efficiency do look impressive, my major concerns are that the MS somewhat suffers from lack of quantifications, fail to describe the overall properties of the cerebral organoids that have high/low efficiencies of generating microglia, as well as the precise description of the protocol that was used to achieve such seemingly highly microglia number within the organoids. The author mentioned briefly in the discussion section that minor modifications were applied to protocol published previously (Lancaster et al 2014), namely Matrigel embedding timing, and microglia numbers were higher under these conditions. However, no direct comparison or additional details were provided in the MS. I think this would be an important point to address since that this is the first report to describe this.

In addition, given the heterogeneity of cerebral organoid (which is totally fine and not necessary represent disadvantage of the system over other published dual-SMAD based systems) it is not clear whether organoid that harbor cortical-like cytoarchitecture or more enriched with neural epithelium tissue also have similar microglia generation efficiency. From what I can tell, both the organoid that expresses/enriched with T gene and microglia markers do not display well-developed neuroepithelium structure that should be presented in optimal organoids at the studies organoids stages.

As outlined below, to address these general concerns we have added quantifications for several of the described assays, performed experiments to establish the overall neuronal properties of the organoids, as well as their content of microglia. Further, immunohistochemical analyses show that the organoids display well-developed neuroepithelium structures. Finally, we provide a better comparison of the protocols used by us and that published in Lancaster et al. (2014).

More specifically:

1. What is the ratio the organoids display microglia-like cells within the same batch and across different batches? Is there variability in the of percentage/number of Iba1+ cells between different organoids? Is there any correlation between batches display more non-neural tissue and microglia differentiation? It would be useful to provide a tile scan of whole sections of organoids stained for neural markers along with Iba1 to show the distribution and to demonstrate that these organoid are indeed neural organoids and not simply neurospheres.

We thank the reviewer for these valuable suggestions and agree that more information on the quantity of IBA-1⁺ cells and the organization of the organoids would further improve the manuscript. To address these suggestions, we have performed additional experiments and have added a new supplementary Fig. 2 in which we quantified the fraction of microglial cells in organoids from the same batch, between batches and from different donors (supplementary Fig. 2a-c). In the methods and results section, we describe how we performed this quantification. Furthermore, we assessed the link between IBA-1 and neuronal quantity

(TUJ1) within these organoids and found that an increase in microglia coincides not with a decrease, but an increase in neuronal tissue (supplementary Fig. 2d). We also show a tiled scan of an organoid in which we assessed neuronal maturation in detail using several different markers PAX6, TUJ1, CTIP2, TBR1 & NEUN. These images show that microglia development in the organoids coincides with proper neuronal maturation and (cortical) organization (supplementary Fig. 2e).

The text that we added is highlighted in the manuscript and shown below.

Results: The fraction IBA-1⁺ cells within organoids showed limited variation between batches, timepoints, and donors (Supplementary Fig. 2a-c), and microglia quantity was positively correlated with TUJ1 expression (Supplementary Fig. 2d). Furthermore, microglia-containing organoids demonstrated proper neuronal development and cortical layer organization, based on expression of NEUN, PAX6, CTIP2, and TBR1 (Supplementary Fig. 2e).” (page 6)

Methods: “The mean percentage of CD11b⁺/CD45⁺ cells of iPSC donors 1, 3, and 5 was 0.83% ± 0.3 at both timepoints for oMG (supplementary Fig. 2c). “(page 22)

“To determine the fraction of TUJ1 and IBA-1 staining in organoid sections, two tiled scans per organoid of three batches from oMG1 were prepared. A macro in FIJI was created with the following steps: background subtraction (rollingball set to 10 (hoechst) or 78 (TUJ1, IBA-1) pixels; contrast was automatically enhanced; automated default threshold applied; particles were quantified with size of > 5 (hoechst) or > 10 (TUJ1, IBA-1) pixels; positive area was summarized; TUJ1 and IBA-1 was normalized by dividing the positive area by the hoechst positive nuclear area (supplementary Fig. 2a, b, d). “ (page 28)

Figures: Supplementary Figure 2. “The quantity of microglia is similar between batches, donors, and timepoints in culture and co-mature with neurons
a- Mask images created by an automated macro in FIJI to quantify the fraction of nuclei, IBA-1, and TUJ1 positive area of a tiled image of a fluorescent staining. The fluorescent channels were split and a separate threshold was applied to enable further analyses. Representative pictures of cerebral organoids from iPSC 1 are shown. Scale bar 500 μm.
b- Quantification of IBA-1/nuclei ratio from tiled fluorescent images of 2 sections per organoid. The variation between batches is similar to the variation between organoids of the same batch ranging from 0.05-0.2.
c- Percentage of CD11b⁺ cells in organoid single cell suspension when sorted with flow cytometry. Organoids from three donors were used for this experiment (iPSC 1, 3, and 5) at two timepoints (38 and 52 days in vitro). The mean percentage of CD11b⁺/CD45⁺ cells of donors iPSC 1, 3, and 5 was 0.83% ± 0.3 (SD) at both time points for oMG (n = 6)
d- The increase of IBA1⁺ is positively correlated with TUJ1. Each data-point reflects the IBA-1 and TUJ1 fraction, normalized to nuclei, of one tiled image of organoids from three separate batches.
e- Neuronal identity and cyto-architecture is maintained in organoids containing microglia as shown in tiled pictures from sections of one organoid after 66 days in culture. Co-staining for: microglia (IBA-1) and mature neurons (NEUN) (* indicates NEUN⁺ cells in close proximity with IBA-1⁺ cells, left panel); radial glia (PAX6) and a pan-neuronal marker (TUJ1, middle panel); and for a deep cortical layer marker (CTIP2) and a post-mitotic projection neuron marker (TBR1, right panel). Scale bar 500 μm.” (page 2-3 supplementary information)

2. What are the modifications from the Lancaster protocol if any? would be good to list those out in the text.

We realize that it would be valuable to the field to know the precise details of our protocol. Therefore, we have included a more detailed protocol describing the generation of organoids. More details were added to the methods section and we have included an additional supplementary Table that step-by-step describes the protocol used by us and that compares these steps to the original protocol of Lancaster et al. (supplementary Table 1)

Methods: "Organoid differentiation

Organoids were differentiated using a modified version of a previously published protocol (supplementary Table 1)¹². In short, after dissociation into single cell suspension with Accutase, 3.5×10^6 cells were seeded per well in an AggreWell800 microwell plate (StemCell Technologies, 27865) in 2 mL of hESC4 (4 ng/mL bFGF) medium supplemented with ROCK-inhibitor Y-27632 (1:100). Approximately 11,500 cells per microwell were left to form embryoid bodies overnight. After 48 hours, embryoid bodies (EB; $325 \pm 40 \mu\text{m}$) were transferred to ultra-low attachment 96-well plates (Corning, 3474). Medium was replaced at day 4 by hESC0 (hESC without bFGF) and at day 6 by neural induction medium (NIM). At day 13, organoids ($570 \pm 50 \mu\text{m}$) were embedded in matrigel (Corning, 356234) and kept in organoid differentiation medium without retinoic acid (RA). Four days later they were transferred to spinning bioreactors on a magnetic platform (27.5 rpm; Pfeiffer, 183-001). No changes were made in the composition of the medium compared to the original protocol with exception of Heparin in the NIM where we used 0.1 $\mu\text{g/mL}$ instead of 1 $\mu\text{g/mL}$. All organoid cultures were qualitatively selected according to the guidelines described previously¹⁷. Seven batches were made from iPSC 1, and 3 batches of iPSC 3 and 5 (Table 1). Each batch had a minimum of 32 organoids." (page 20-21)

Figures/tables: "Supplementary Table 1. Comparative overview of adaptations in the organoid differentiation protocol used in this study and the original protocol of Lancaster et al. 2014." (page 11 of supplementary information)

Discussion: "... we reduced levels of neuroectoderm stimulant heparin and delayed the matrigel embedment of the organoids." (page 14)

3. Figure 3: What genes are driving the differences between the clustering? What's the difference between adult MG and oMG?

As suggested by the reviewer we further explored the genes that drive the clustering. As we found sex-specific genes in the initial top gene lists, we first corrected our data for sex. We re-ran the analyses, identified up- and down-regulated genes and additionally performed a panther GO-slim functional annotation on the top 100 differentially expressed genes to identify associated pathways. Most of the associated pathways overlapped between adult MG and oMG. These extra analyses are added to the results section.

Results: "To further analyze differences in gene expression between oMG and adult MG, we analyzed transcriptomic differences between day 52 oMG and adult MG1 (supplementary Fig. 4f-h). 1220 genes were significantly upregulated in oMG and 1271 genes in MG1 (FDR < 0.05). Top 100 upregulated genes in oMG included FNI, CDH4, COL1A2 and COL3A1 for oMG and, CDH1, CXCR1 and ACKR2 for adult MG1 (supplementary Table 2). Functional gene classification with Panther GO-Slim²⁷ on the top 100 upregulated genes in both day 52 oMG and adult MG showed that the significantly upregulated genes in both samples belong to similar biological processes suggesting only subtle differences between oMG and adult MG. GO-pathway analysis on all DEGs did not show any significantly enriched pathways in adult MG1 versus oMG, whereas oMG showed enrichment for "collagen catabolic process", "extracellular matrix organization", "negative chemotaxis", "positive regulation of synapse assembly" and "endodermal cell differentiation" compared to adult MG1 (supplementary Table 3). Myeloid function-related terms were detected by GO-pathway analysis of common

genes (between adult MG1 and day 52 oMG), such as “nervous system development”, “interleukin-12-mediated signaling pathway” and “neutrophil degranulation” (supplementary Table 3).” (page 9)

4. It would be important to provide some kind of quantification of the levels phagocytosis following exposure of oMG to iC3b-coated beads in Fig 4.

To address this point, we have performed new phagocytosis assays on adult MG and oMG followed by quantification of the number of internalized beads/cell after 30 minutes and 1 hour using microscopy. As expected, internalization of beads was increased in time. We present these new data in the Results section and in Figure 4f. In addition, the methods section was expanded to further detail this assay.

Results: “To test phagocytic capacity, we exposed oMG day 79 and adult MG to iC3b-coated beads for 0.5 and 1 hr. Beads were detected in adult MG and oMG and as expected the number of phagocytosed beads increased in time (Figure 4e, f). This suggests that phagocytosis regulated via the C3 receptor is functional in oMG.” (page 11)

Methods: “Bead engulfment by adult MG and oMG was analyzed on three randomly selected view fields per section and per condition (oMG1, for 0.5 and 1 hr; adult MG1.7, 1.8, and 1.9, for 0.5 and 1hr). IBA-1⁺ cells were categorized based on the number of inoculated beads as follows: 0 (type 1), 1-5 (type 2), 6-15 (type 3), or > 15 (type 4) using FIJI version 1.49 software.” (page 26)

Figure:4 f-f- Quantification of iC3b beads engulfment by oMG and adult MG after 0.5 and 1 hr exposure to the beads. Three randomly selected view fields per condition were manually quantified (oMG 1, for 0.5 and 1 hr; adult MG1.7, 1.8, and 1.9, for 0.5 and 1 hr). IBA-1⁺ cells were categorized based on the number of inoculated beads as follows: 0 (type 1), 1-5 (type 2), 6-15 (type 3), or > 15 (type 4).” (page 41)

5. In Fig. 6, LPS stimulation was applied for 24h, relative to 6h in oMG cultured in vitro. Does the inflammation response occur mainly after 24h in this experiment? Or similar induction can be achieved after 6h? Also, the majority of experiments showed low levels of cytokines induction. This can be explained by different levels of oMG within the organoids or alternatively low reactivity. This point can be addressed by correlation of the secretion/expression levels of cytokines with the average number of microglia cells in the organoids to clarify this point.

To address this point additional information was added to the methods section explaining that a pilot experiment was performed to determine the optimal timing for cytokine secretion assessment. We found that the mRNA levels peak at 6 hours, but that secretion only became significantly detectable 24 hr after LPS stimulation (due to limited space in the manuscript we added the figure below). Regarding the low increase in cytokine induction, the reviewer suggests that one explanation could be low reactivity. In our opinion these lower values are the result of the relatively low number of microglia in an organoid (~10.000) and the large, and required, volume of medium in which they are kept. This is also in accordance with data from the stimulations in isolated microglia, with approximately 10x more microglia cells in ¼ of the medium, where values are closer to what is normally observed. We now discuss and address this issue in the results section.

Figure. Cytokine secretion following LPS stimulation. The cytokine response was only significantly increased after 24 hr determined with a Friedman's ANOVA test ($p = 0.03$; Dunn's test $t = 6$ versus baseline: -3 with $p = 0.44$; Dunn's test $t = 24$ versus baseline: -6 with $p = 0.03$).

Results: "The cytokine response to LPS detected in the medium was low as compared to assays performed with isolated oMG. However, this is expected considering that only ~1% of all cells of an organoid are microglia (~10,000 cells) and that 100,000 cells were used in the 2D experiments." (page 13)

Methods: "A pilot experiment to determine the optimal timing for cytokine measurements, showed that the optimal timing for cytokine secretion following LPS stimulation is 24 hr. Therefore, we used 24 hr to measure the acute response, and 72 hr for the effect on the long-term." (page 29)

6. The upregulation of adult microglia transcriptome signature in oMG is impressive, however, from what I can tell, oMG lack/has a very low expression of key markers of adult microglia, for example, TMEM119, P2RY12, and TREM2. It would be important though to discuss this point. One would expect that prolonged culture of organoids beyond day 52 may achieve improve maturity. I am not necessarily suggesting analysis of transcriptome beyond day 52, but this point could be addressed by qRT-PCR/immunostaining for these markers.

We followed the reviewer's suggestion to analyse the expression of key microglia genes beyond day 52 in culture. We sorted microglia at an age of 119 days in culture and compared gene expression using RT-PCR for several key microglia genes (including the suggested TMEM119, P2RY12 and TREM2) with the 52 days in culture timepoint. We indeed detected an increase in the expression of the key markers we investigated in time. We now made separate plots of these key genes selected from the RNAseq data to show these differences between adult MG and oMG (supplementary Fig. 4b) and added plots with the RT-PCR data of these genes between oMG 52 and oMG 119 (supplementary Fig. 4c), as also described in the results and discussion section.

Results: "We found that expression of part of these genes, like AIF1 and RUNX1, is comparable between oMG and MG1, but for some the levels are 3-fold (PTPRC or CX3CR1) or even 10-100-fold different (TREM2, P2RY12, TMEM119) (supplemental Fig. 4b). To analyze whether microglia continued to mature within the organoid, we also analyzed the expression of these markers at 119 days in culture. Microglia from iPSC donors 1, 3, and 5 at day 119 versus day 52 showed an increase in expression of all these selected typical microglia genes AIF1, RUNX1, PTPRC, CX3CR1, TREM2, P2RY12, and TMEM119 (supplemental Fig. 4c)." (page 7-8)

Discussion: “Some key genes like P2RY12, TREM2, and TMEM119 that were lowly expressed in oMG cultured for 52 days compared to adult MG, increased when oMG were held in culture longer (119 days).” (page 16)

Figure: supplementary Fig. 4g, h” g- Plotted DESeq2 normalized expression of selection of microglia typical genes in oMG day 52 and adult MG1.

h- Overtime mRNA expression of characteristic microglia markers. Microglia were enriched with CD11b-MACs from organoids after 52 or 119 days in culture. mRNA levels were assessed by qRT-PCR and normalized to the geomean of the reference genes SDHA2 and ACTB. Data points represent mRNA levels of oMG 1, 3 and 5. “ (page 4-5)

7. The similarity between the oMG and adult MG in Fig3 is impressive, however, the difference between fetal and adult MG is only 0.37 which is quite surprising. Such difference can be partially explained by the fact that datasets were obtained from different resources and also can depend on the methods that the data were normalized, it would be important to explain and discuss this point.

It is indeed surprising that the correlation between fetal and adult MG is small. We now mention this low correlation in the results section and discuss it in the discussion.

Results: “The correlation between adult MG1 and fetal MG was low ($\rho = 0.37$) in this analysis (Figure 3e).” (page 8)

Discussion: “When comparing a selection of transcription factors that are important for microglia in vivo²⁶, adult MG had the lowest correlation with fetal MG which could reflect a maturation effect. Simultaneously, oMG strongly correlated with adult MG whereas fetal MG correlated with iPSC MG. Still, comparisons between multiple available datasets, including the ones in this study, have inherent limitations due to differences in sample processing of the different resources.” (page 16)

Minor point:

1. Line 100, reference 12 should be reference 17 to refer to the original study.

We thank the reviewer for this remark, we corrected the manuscript accordingly.

Reviewer #2

This manuscript by Ormel et al. describes a new cerebral organoid model in which microglial cells are found. These cells have a ramified morphology, a transcriptomic profile resembling mature microglia isolated from human postmortem brain tissue, and may also engulf synaptic materials as observed during normal physiological conditions. To my knowledge this is the first report of microglia spontaneously developing within cerebral organoids, which makes it novel and interesting to a broad readership.

We thank the reviewer for this positive assessment.

Considering my expertise I will comment on the morphological aspects of this study, for which I have a few concerns as detailed below. In particular, without methodological information it is very difficult to assess whether the analyses of microglial morphology, phagocytosis, and synaptic pruning were properly conducted. It would be useful in the Suppl. materials to provide information pertaining to the imaging and analysis itself.

We agree with the reviewer that it is important that we present the methods as detailed as possible to enable future research on this subject. In the revised manuscript, different

analyses are now described in more detail, including descriptions of new experiments related to phagocytosis and STED microscopy.

Methods: “Quantification of immunohistochemistry

Microglia perimeter was quantified using a macro in FIJI software on sections of four separate organoids per timepoint. The macro consisted of the following steps: transformation to 8-bit; scale set to μm ; automated default threshold applied; converted to mask (particles filtered with size $> 40 \mu\text{m}$; option “outline”; the outline area was summarized and averaged over total number of microglia in the image to obtain the perimeter/microglia cell (figure 1f). To determine the fraction of TUJ1 and IBA-1 staining in organoid sections, two tiled scans per organoid of three batches from oMG1 were prepared. A macro in FIJI was created with the following steps: background subtraction (rollingball set to 10 (hoechst) or 78 (TUJ1, IBA-1) pixels; contrast was automatically enhanced; automated default threshold applied; particles were quantified with size of > 5 (hoechst) or > 10 (TUJ1, IBA-1) pixels; positive area was summarized; TUJ1 and IBA-1 was normalized by dividing the positive area by the hoechst positive nuclear area (supplementary Fig. 2a, b, d). “ (page 28)

Regarding the analysis of synaptic pruning, the complete engulfment of PSD-95 by IBA1-positive cells is not sufficiently convincing in my opinion, considering that confocal microscopy does not have the resolution required to discriminate between partial versus complete microglial engulfments. These findings should be strengthened. In particular, in Fig. 5b, the PSD-95 puncta appears to be encircled rather than engulfed. Recently, different mechanisms underlying pruning have emerged: synaptic stripping and phagocytosis, but also trogocytosis and extracellular digestion. Synaptic stripping is the physical separation of pre- and post-synaptic elements by microglial processes. Phagocytosis is their complete internalisation within endosomes fusing with lysosomes. Trogocytosis involves the removal of small pieces of presynaptic boutons rather than their engulfment as a whole, while extracellular digestion refers to the degradation of materials inside the extracellular space. Additionally, it would be useful to assess whether microglia in the organoids also interact with presynaptic boutons, in addition to dendritic structures.

We agree with the reviewer that confocal microscopy does not provide enough resolution to unequivocally show that PSD-95-positive material is completely engulfed by oMG (or not). We apologize if we gave the impression that we could. This section of our manuscript was intended to probe future applications for the oMG/organoid model without any claim that we unequivocally determined whether or how oMG regulate synaptic pruning.

However, to address the reviewer’s concern we have now collaborated with the group of Harold MacGillavry (Utrecht University) who are experts on super-resolution stimulation emission depletion (STED) microscopy. STED does provide the resolution to determine whether PSD-95 material would be engulfed (or not) by oMG. New STED data is added to the results section showing that there is indeed engulfment of PSD-95 particles by microglia (Figure 5b). Interestingly, while some PSD-95 particles are clearly inside microglia, others are in close proximity or partially engulfed. This most likely indicates that different mechanisms of synaptic pruning may be at play in organoids, as the reviewer indicates.

We also thank the reviewer for pointing out the mechanisms for synaptic pruning that have recently been described (which we have included in the manuscript). Our manuscript is mainly focused on identifying and characterizing oMG in cerebral organoids, and in the final section of the manuscript we probe two future applications of this model (synaptic function and inflammation). It will be very interesting to study if and how oMG regulate synaptic pruning, and we have extended the discussion on this topic in our revised manuscript. However, answering this question, would require a new set of tools (e.g. life cell imaging of organoids at high resolution, with microglia and neurons expressing fluorescent reporters). We feel that, albeit interesting, such studies are outside of the scope of our current manuscript.

Results: “These data suggest that cerebral organoids contain functional synapses and could be used for studying the potential effect of microglia on synapse function. But can oMG processes be found in close proximity to neuronal synaptic structures? To address this question, we applied super-resolution stimulated emission depletion (STED) microscopy to study the distribution of microglia (IBA-1) in relation to the post-synaptic marker PSD-95 in cerebral organoids. High-resolution images from the STED experiment showed significant overlap between IBA-1 and PSD-95 signals, with PSD-95 material either inside IBA-1⁺ oMG processes or being in contact with or partially engulfed by oMG ramifications (Figure 5b).” (page 12)

*Methods: “Super-resolution STED microscopy acquisition and image processing
Dual-color gated STED (gSTED) imaging of day 66 organoids immunostained for IBA-1 and PSD-95 was performed with a Leica TCS SP8 STED 3X microscope using a HC PL APO 100x/ 1.4 oil immersion STED WHITE objective. The 590 nm and 647 nm wavelengths of pulsed white laser (80MHz) were used to excite the Alexa594-labeled PSD-95 and the Atto647N-labeled IBA-1, respectively. Both Alexa594 and Atto647N were depleted with the 775 nm pulsed depletion laser (10%-20% of maximum power) and we used an internal Leica HyD hybrid detector (set at 100% gain) with a time gate of $0.3 \leq t_g \leq 6$ ns. Multiple Z-stack were obtained at 0.18 μ m interval to acquire 4.0 μ m image stacks (corrected for refractive index mismatch between oil immersion objective and FluorSave, the mounting medium used) in 2D STED mode using the 100x objective with ~ 2 x zoom to obtain approximately 80 nm pixel size. Raw gSTED images collected using Leica TCS SP8 were subjected to deconvolution as stacks using Huygens deconvolution software. Deconvolution of the 594 and 647 channels was performed separately using the CMLE deconvolution algorithm, with a maximum of 40 iterations and the signal-to-noise ratio (SNR) set at 7.” (page 27-28)*

Discussion: “Functional synapses were detected in the organoids and microscopy-based approaches detected microglial ramifications in close proximity to neuronal processes and synaptic material. Different mechanisms of synaptic pruning have been described, including synaptic stripping, phagocytosis, and trogocytosis^{29,36}. Our super-resolution microscopy data detected post-synaptic material inside and in close proximity to oMG, suggesting these different mechanisms may be at play as has been reported for MG. Future studies are needed to dissect these mechanisms for oMG in cerebral organoids, but our data further identify the organoid model as a valuable tool for studying human microglia-neuron interactions in 3D brain tissue.” (page 15-16)

*Figure: Figure 5” b- gSTED microscopy showing synaptic content inside and partially engulfed (indicated by *) by microglial processes visualized by immunohistochemistry for IBA-1 and PSD-95 on day 66 organoids. Maximum intensity projection of the entire cell (top, scale bar 10 μ m), and close-up of region of interest (box in dashed line) with maximum intensity projection (bottom, scale bar 1 μ m) and orthogonal view (voxel size 0.18 μ m). “ (page 41)*

REVIEWERS' COMMENTS:

Reviewer #1 (Remarks to the Author):

The manuscript by Ormel et al. has significantly improved since the last version. The addition of quantifications for the fraction of microglia-like cells in the organoids across batches and the phagocytosis quantifications are very helpful, as is the addition of further neuronal markers as well as the tiled scan of the entire organoid section. The upregulation in the expression of key microglia markers following organoid culture beyond day 52 is also quite interesting and suggests progressive maturation of oMG over time. Overall, with the additions now provided, the experimental work seems solid. However, there are still a few issues with some analyses and the presentation of the data that needs some work. These are detailed below:

- 1- The extended analysis of gene expression differences between oMG and adult MG is very informative, but is however not presented in the most informative way. It would be more informative to present the stats of differentially expressed genes in a plot, for example a volcano plot. In addition, I have concerns about the clustering analysis. From what I can tell, the fib/iPS were sequenced using a different paradigm as compared to the other MG samples (paired vs single end sequencing) and on different sequencing platforms. As this is not an issue per se, it would be important to either compute/separate a potential batch effect or to at least perform hierarchical clustering without Fib/iPS and present the data separately.
- 2- The upregulation of key MG genes after longer culture of organoids is a significant finding and I suggest to move it to a main figure.
- 3- Methods: The final concentration of Rock1 should be used rather than the dilution.
- 4- The graph that describes the pilot experiment to determine the optimal timing for cytokine secretion (6h vs 24hrs) is informative and should be added to the manuscript.
- 5- Line 372: developed (in) the past decade to study specific cell

Reviewer #2 (Remarks to the Author):

The revision has addressed all of my concerns.

Point-by-point response to referees letter

Reviewer #1

The manuscript by Ormel et al. has significantly improved since the last version. The addition of quantifications for the fraction of microglia-like cells in the organoids across batches and the phagocytosis quantifications are very helpful, as is the addition of further neuronal markers as well as the tiled scan of the entire organoid section. The upregulation in the expression of key microglia markers following organoid culture beyond day 52 is also quite interesting and suggests progressive maturation of oMG over time. Overall, with the additions now provided, the experimental work seems solid. However, there are still a few issues with some analyses and the presentation of the data that needs some work. These are detailed below:

We thank the reviewer for the positive assessment of the revisions made. As outlined below, we have addressed the remaining concerns of this reviewer.

1- The extended analysis of gene expression differences between oMG and adult MG is very informative, but is however not presented in the most informative way. It would be more informative to present the stats of differentially expressed genes in a plot, for example a volcano plot. In addition, I have concerns about the clustering analysis. From what I can tell, the fib/iPS were sequenced using a different paradigm as compared to the other MG samples (paired vs single end sequencing) and on different sequencing platforms. As this is not an issue per se, it would be important to either compute/separate a potential batch effect or to at least perform hierarchical clustering without Fib/iPS and present the data separately.

To address this concern, we have followed the reviewer's suggestions and added volcano plots to the manuscript displaying the differentially expressed genes. We display the DEG data in volcano plots prior to the Log2fold change correction to best illustrate the data and use it to explain to the reader why we choose to perform this correction.

Supplementary Fig. 4

f, g, and h- Volcano plots show differentially expressed genes (FDR < 0.05 in red) between day 38 oMG vs adult MG1 (f), day 52 vs day 38 oMG (g), and day 52 oMG vs adult MG1 (h). (Supplementary information, page 9)

Results: Unsupervised hierarchical clustering shows that oMG clustered separately from adult microglia (adult MG1) (Supplementary Fig. 4b). When comparing the data with the original iPSC lines and fibroblasts from the same donors, unsupervised hierarchical clustering, Spearman correlation coefficients, and principal component analyses (PCA) (Figure 3a, b; Supplementary Fig. 4c) showed that both at day 38 and day 52 oMG closely resemble adult MG1. (page 7)

Furthermore, we have added a separate unsupervised hierarchical cluster analysis solely performed on the oMG and adult MG samples to address the reviewer's concerns regarding batch effects.

Supplementary Fig. 4 b- Unsupervised hierarchical cluster analysis on DESeq2 rlog transformed raw counts of oMG day 38, oMG day 52, and adult MG1 based on all genes after removal of common genes (FDR > 0.05, sum of raw read counts > 0) between samples. (supplementary information page 9)

2- The upregulation of key MG genes after longer culture of organoids is a significant finding and I suggest to move it to a main figure.

We moved the plots to Figure 3.

3- Methods: The final concentration of Rocki should be used rather than the dilution.

We have added the final concentration of Rocki.

4- The graph that describes the pilot experiment to determine the optimal timing for cytokine secretion (6h vs 24hrs) is informative and should be added to the manuscript.

We have added the graph to Figure 5 and describe the pilot experiment in the methods.

Methods: A pilot experiment to determine the optimal timing for cytokine measurements, showed that 24 hr after LPS stimulation the cytokine response was significantly increased as determined with a Friedman's ANOVA test ($p = 0.03$; Dunn's test $t = 6$ versus baseline: -3 with $p = 0.44$; Dunn's test $t = 24$ versus baseline: -6 with $p = 0.03$) (Figure 5c). (page 29)

Figure 5c: c- Pilot experiment in which the IL6 inflammatory response was measured in organoids in situ. The cytokine response was significantly increased after 24 hr as determined with a Friedman's ANOVA test ($p = 0.03$; Dunn's test $t = 6$ versus baseline: -3 with $p = 0.44$; Dunn's test $t = 24$ versus baseline: -6 with $p = 0.03$). $N = 3$ iPSC 1 organoids per group. Error bars indicate standard deviation (page 42)

5- Line 372: developed (in) the past decade to study specific cell

We have corrected this mistake.

Reviewer #2

The revision has addressed all of my concerns.

We thank the reviewer for this positive assessment